# RING domains act as both substrate and enzyme in a catalytic arrangement to drive self-anchored ubiquitination

Leo Kiss [1✉], Dean Clift [1], Nadine Renner[1], David Neuhaus [1] & Leo C. James [1✉]

Attachment of ubiquitin (Ub) to proteins is one of the most abundant and versatile of all posttranslational modifications and affects outcomes in essentially all physiological processes. RING E3 ligases target E2 Ub-conjugating enzymes to the substrate, resulting in its ubiquitination. However, the mechanism by which a ubiquitin chain is formed on the substrate remains elusive. Here we demonstrate how substrate binding can induce a specific RING topology that enables self-ubiquitination. By analyzing a catalytically trapped structure showing the initiation of TRIM21 RING-anchored ubiquitin chain elongation, and in combination with a kinetic study, we illuminate the chemical mechanism of ubiquitin conjugation. Moreover, biochemical and cellular experiments show that the topology found in the structure can be induced by substrate binding. Our results provide insights into ubiquitin chain formation on a structural, biochemical and cellular level with broad implications for targeted protein degradation.

[1] MRC Laboratory of Molecular Biology, Cambridge, UK. ✉email: lkiss@mrc-lmb.cam.ac.uk; lcj@mrc-lmb.cam.ac.uk

Ubiquitin conjugation modifies cellular proteins with a highly versatile label and is used in virtually all physiological pathways. The huge variety of possible ubiquitination patterns includes modification with single ubiquitin molecules and/or up to eight different ubiquitin chain types[1]. Furthermore, such modifications can be combined resulting in highly specific signals that trigger dedicated processes. These modifications are achieved by a three-enzyme cascade. Ubiquitin is first activated by an E1 enzyme and charged onto the active site cysteine of an E2 Ub-conjugating enzyme. Then the E2~Ub and a third enzyme, an E3 ligase, ubiquitinate the substrate. Structural insights into the process of substrate ubiquitination are limited to the transfer of one ubiquitin[2] or ubiquitin-like protein[3,4]. How specificity is achieved within the ubiquitination system remains largely elusive due to its extremely high level of complexity.

Ubiquitin chains that are linked via lysine 63 (K63) are involved in endocytosis, DNA damage, and the immune response[1]. The only E2 enzyme dedicated to K63-ubiquitination is Ube2N (Ubc13), which forms a heterodimer with either Ube2V2 (Mms2) or Ube2V1 (Uev1)[5–7]. Ube2V2 binds and orients the acceptor ubiquitin, thereby generating specificity for K63 linkage[8–10]. While Ube2N/Ube2V2 efficiently forms free K63-linked ubiquitin chains, substrate modification requires the presence of a priming ubiquitin for elongation[11]. Such a mechanism has been established for the RING (really interesting new gene) E3s TRIM21 and TRIM5α[12,13]. With ~100 members in humans, the TRIM family contains the largest number of distinct RING E3s[14]. TRIM21 detects antibody-coated viruses by specifically recognizing the bound antibody. This activates its E3 ligase function, resulting in virus neutralization and innate immune activation via the NF-κB pathway[15,16]. First, Ube2W attaches a priming ubiquitin to the TRIM21 N-terminus[13]. Next, TRIM21 specifically recruits Ube2N/Ube2V2 to produce a TRIM21-anchored K63-linked ubiquitin chain[13,17]. TRIM21 does not engage its target directly and forms the ubiquitin chain on its own N-terminus, resulting in degradation of itself, the antibody, and the substrate[13,15]. This mechanism can be repurposed by introduction of substrate-specific antibodies to the cytosol, resulting in targeted protein degradation by Trim-Away[18]. This is conceptually similar to proteolysis targeting chimeras (PROTACs), that bring substrate and E3 ligase together, resulting in substrate ubiquitination and degradation[19].

In this work, we sought to understand how RING E3-anchored ubiquitin chains are formed. Although it is known how Ube2N/Ube2V2 specifically connects two ubiquitin molecules with a K63 linkage[8–10] and how Ube2N is specifically recruited by RING E3s[17], it was not known how substrate-anchored ubiquitin chains are formed. Yet, anchored ubiquitin chains deliver highly specialized functions and phenotypes. We address this question by solving a structure trapped in the process of RING-anchored chain elongation. Together with a kinetic analysis, this reveals how chemical activation of the acceptor ubiquitin is achieved. It also reveals a specific topology of the RING domains that is necessary for formation of the self-anchored ubiquitin chain. Finally, we validate the importance of this arrangement biochemically and show that it results in targeted protein degradation in a physiological setting.

## Results

### Structure showing the formation of anchored ubiquitin chains.
We set out to understand how a substrate-bound ubiquitin chain can be formed. In principle, ubiquitin chain elongation of TRIM proteins depends on their RING domain only. In the case of TRIM21 (and TRIM5α), the TRIM RING itself is the substrate, after it has undergone N-terminal mono-ubiquitination upon interaction with the E2 enzyme Ube2W[12,13,20]. Therefore, we attempted to address substrate-bound ubiquitination with TRIM21 RING and its chain forming E2 heterodimer Ube2N/Ube2V2. In crystallization trials, we used N-terminally mono-ubiquitinated TRIM21 RING domain ($Ub^{G75/76A}$-$TRIM21^{1–85}$ or Ub-R), an isopeptide-linked, non-hydrolyzable ubiquitin-charged Ube2N conjugate (Ube2N~Ub) and Ube2V2. We solved the atomic structure of this complex at 2.2 Å resolution, with one copy each of Ub-R, Ube2N~Ub and Ube2V2 in the asymmetric unit (Supplementary Fig. 1 and Supplementary Table 1). The naturally occurring TRIM21 RING homodimer[21] was generated in our model by invoking crystal symmetry (Fig. 1a). The RINGs engage Ube2N~Ub in the closed conformation[22–24] and Ube2N forms a heterodimer with Ube2V2 (refs. [5–7]). Analyzing further interactions within the crystal lattice, we found that the TRIM21-linked ubiquitin made additional contacts to Ube2N/Ube2V2 of a symmetry-related complex (Fig. 1b and Supplementary Fig. 1), which orient the RING-bound ubiquitin so that its K63 points toward the active site, ready for nucleophilic attack (Fig. 1b, c). Our structure thus represents a snapshot of a ubiquitin-primed RING ready for self-anchored ubiquitin chain elongation.

### Chemical mechanism of ubiquitination.
Having captured a 2.2 Å resolution representation of the system prior to catalysis, we were able to perform a detailed analysis of ubiquitin transfer. The Ube2N-charged ubiquitin can be found in the RING-promoted closed Ube2N~Ub conformation and thus represents the donor ubiquitin (Fig. 1). The RING-bound ubiquitin of a symmetry-related complex was captured by Ube2N/Ube2V2, positioning its nucleophilic K63 $N^ζH_3$ group 4.8 Å from the electrophilic carbonyl of the donor ubiquitin C-terminus (Fig. 2a and Supplementary Fig. 1). Interestingly, K63 of this acceptor ubiquitin shows a direct interaction with D119 of Ube2N (Fig. 2a). This suggests that D119 deprotonates K63 on the acceptor ubiquitin, thereby activating it for nucleophilic attack. Indeed, the corresponding residue in Ube2D (D117) has been suggested to be involved in positioning and/or activating an incoming acceptor lysine[22].

To investigate the chemical mechanism of ubiquitination (Fig. 2b), we measured the kinetics of di-ubiquitin formation (Supplementary Fig. 2). The acid coefficient ($pK_a$) of this reaction should solely depend on the protonation state of its nucleophile, K63. Fitting the ubiquitination velocity of reactions carried out at different pHs to an equation assuming one titratable group revealed a $pK_a$ of 8.3 for Ube2N (Fig. 2c and Supplementary Fig. 2), comparable to what was observed for the SUMO-E2 Ube2I[25]. This is significantly lower than the $pK_a$ of 10.5 for a free lysine ζ-amino group[26], which would be incompatible with catalysis at physiological pH ~7.34 (ref. [27]). We mutated D119 to either alanine or asparagine as neither can act as a base, but asparagine could still bind and orient K63. Both mutants increased the $pK_a$ to ~9 (Fig. 2c). At physiological pH, Ube2N$^{D119A/N}$ modestly increased the $K_M$ by ~4- and ~7-fold, respectively (Fig. 2d). Mutation to alanine reduced $k_{cat}$ 100-fold and mutation to asparagine 30-fold (Fig. 2e), suggesting that substrate turnover also depends on orientation of the lysine nucleophile. Yet, this catalytic rate does not yield efficient ubiquitin chain formation under physiological pH (Supplementary Fig. 3). Together, these observations establish that D119 is the base that deprotonates the incoming acceptor lysine to enable catalysis.

Interactions between ubiquitin and other proteins have been shown to depend on specific conformations of ubiquitin's $β_1$–$β_2$ loop, which can be found in either loop-in or loop-out conformations[28]. These motions change the ubiquitin core

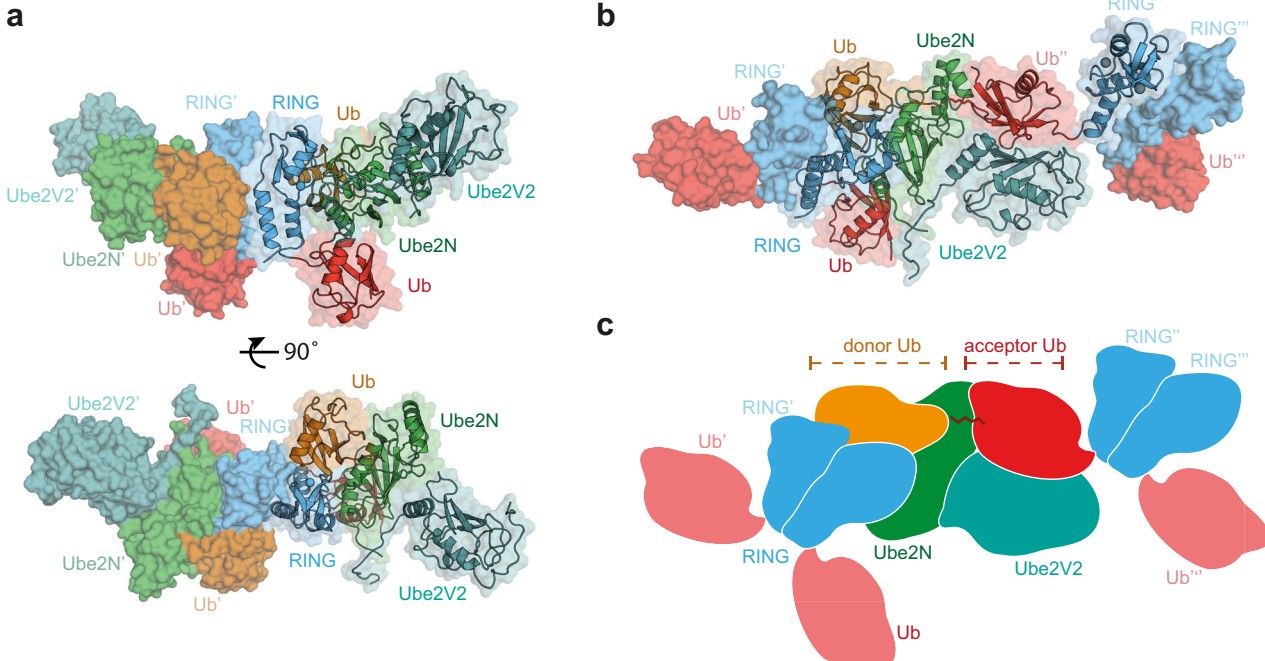

**Fig. 1 Structure of initiation of RING-anchored ubiquitin chain elongation. a** Side and top view of the Ub-R:Ube2N~Ub:Ube2V2 structure (Ub-R, Ub in red, R in blue, Ube2N~Ub, Ube2N in green, Ub in orange, and Ube2V2 in teal). Chains drawn as cartoon represent the asymmetric unit. **b** The canonical model of initiation of RING-anchored ubiquitin chain elongation. **c** Schematic cartoon, representing the canonical model of RING-anchored ubiquitin chain elongation shown in **b**. Symmetry mates are denoted by ′ next to the label. Ub ubiquitin, Ub-R ubiquitin-RING.

structure and subsequent conformational selection enables ubiquitin to interact with many different binding partners[29]. In our structure, we found the donor ubiquitin $\beta_1$–$\beta_2$ loop in its loop-in configuration, and loop-out to be incompatible with formation of the closed conformation (Supplementary Fig. 4a, b). Conversely, the acceptor ubiquitin was in a loop-out configuration (Supplementary Fig. 4c), which appears to be the default state in ubiquitin[28]. Donor and acceptor ubiquitin also have distinct *B*-factor profiles (Supplementary Fig. 4d), perhaps reflecting some other aspect of their different roles in catalysis. Interestingly, the $\beta_1$–$\beta_2$ loop conformation also appears to be critical in ubiquitin-like proteins, such as Nedd8, when activating cullin-RING-ligases (CRL)[2].

RING E3s act by locking the normally very dynamic E2~Ub species in a closed conformation, thereby priming it for catalysis (Fig. 2a)[22–24]. Comparison with our previously determined TRIM21 R:Ube2N~Ub structure[17] shows scarcely any difference between the donor ubiquitin C-termini and the Ube2N active site (Supplementary Fig. 5a). Nonetheless, formation of the closed Ube2N~Ub conformation alone is not sufficient for catalysis, as this also requires the presence of Ube2V2 (ref. [5]), which binds and orients the acceptor ubiquitin[8–10]. We gained additional insight into how Ube2V2 positions the acceptor ubiquitin by analyzing a Ube2N~Ub:Ube2V2 complex that we solved at 2.5 Å resolution (Supplementary Fig. 6 and Supplementary Table 1). By invoking crystal symmetry, this structure shows the orientation of an acceptor ubiquitin by Ube2V2, so that its K63 is pointed toward the active site of Ube2N (Supplementary Fig. 6), an orientation comparable with a structure of yeast Ube2N~Ub:Ube2V2 that was solved in a different crystal lattice[9]. Without a RING present, the donor ubiquitin is not in the closed conformation and our Ube2N~Ub: Ube2V2 structure thus represents an inactive complex. Alignment to our Ub-R:Ube2N~Ub:Ube2V2 structure (Supplementary Fig. 5b–d) reveals that Ube2N and Ube2V2 are packed more closely against each other, resulting in additional contacts

between the acceptor ubiquitin and Ube2N (Fig. 2a and Supplementary Fig. 5c), that position the nucleophile K63 much nearer to the active site (4.8 vs. 7.5 Å). This is achieved because Ube2N N123 and D124 contact ubiquitin via the amide of ubiquitin K63 and the sidechains of S57 and Q62, respectively (Fig. 2a). The ~3-fold reduction in $k_{cat}$ (Fig. 2e) for the mutants Ube2N[N123A] and Ube2N[D124A] suggest that the function of these residues is to finetune the ubiquitination reaction by aiding orientation of the nucleophile. Taken together, the features of our structure trapped in the process of ubiquitin chain formation provide mechanistic insight into how the RING E3 promotes catalysis, by simultaneously activating Ube2N for ubiquitin discharge and allowing Ube2V2 to precisely orient the acceptor ubiquitin.

**The mechanism of RING-anchored ubiquitination.** Next, we sought to understand how RING-anchored ubiquitin chains are formed. In our crystal structure, one RING dimer is positioned so as to mediate the elongation of another mono-ubiquitinated RING in *trans* (Figs. 1b, c and 3a). Importantly, this mechanism depends only on binding of the RING-anchored acceptor ubiquitin to Ube2N/Ube2V2, as no contacts with the RING itself could be observed in our crystal structure (Supplementary Fig. 1). The relative topology of the different RING domains (enzyme and substrate) is thus mostly dictated by the catalytic interfaces, resulting in a ~9 nm separation between the enzyme and substrate RINGs (Fig. 3a). We refer to this arrangement as the catalytic RING topology, in which a RING dimer acts as an enzyme and at least one further RING acts as the substrate for ubiquitination. This topology is not rigid since the linkers between the acceptor ubiquitin and the RING (~3 nm apart), and the RING and the next (B-box) domain in the TRIM ligase (~3.5 nm apart) likely provide additional flexibility (Fig. 3a, b and Supplementary Fig. 1c). In our structure, it is clear that initiation of TRIM21-anchored chain elongation cannot occur in *cis*, as the priming

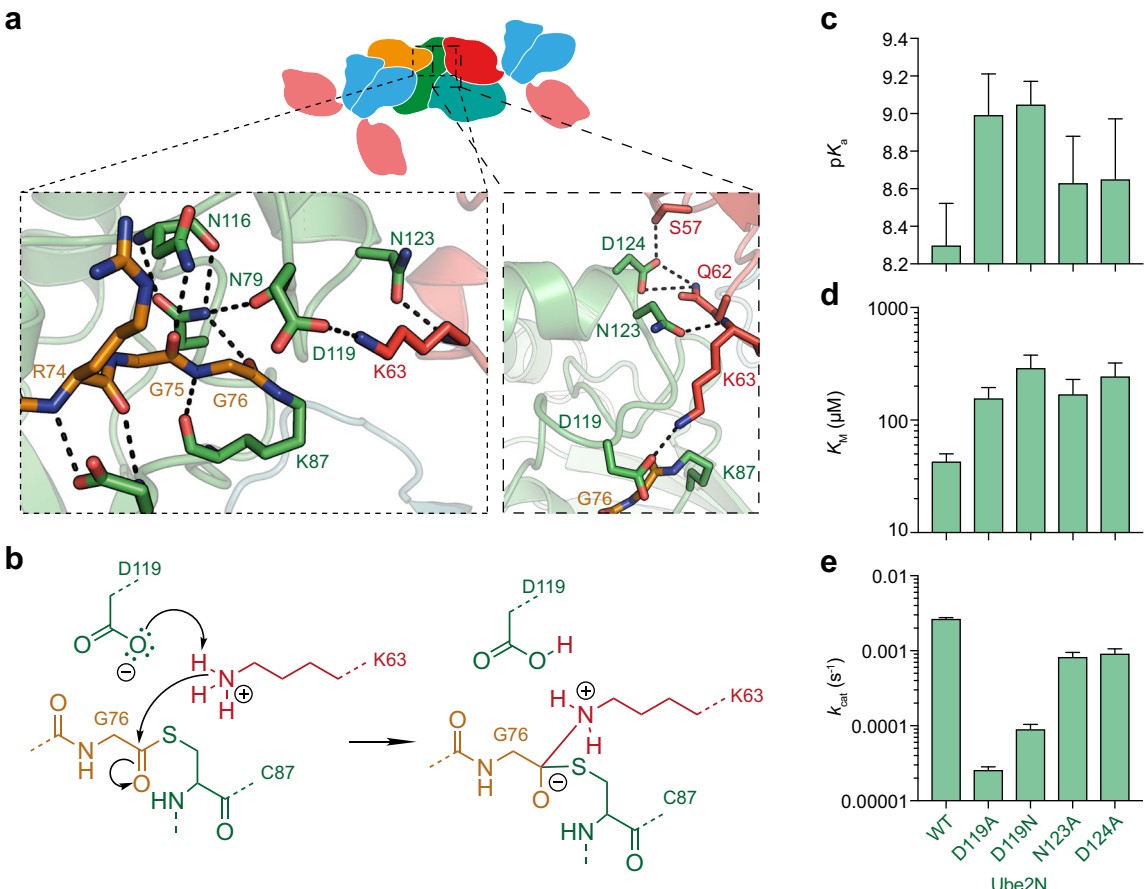

**Fig. 2 Chemical mechanism of ubiquitination. a** Magnified regions of the active site of Ube2N~Ub/Ube2V2 (Ube2N in green, donor Ub in orange, acceptor Ub in red, and Ube2V2 in teal). Stereo-images are shown in Supplementary Fig. 1b. **b** Chemical scheme for the activation of the acceptor lysine. **c** Acid coefficients (p$K_a$), **d** $K_M$, and **e** $k_{cat}$ of di-ubiquitin formation by Ube2N/V2 are presented as best fit + standard error. Fits were performed using data of $n = 3$ technical replicates, and are shown in Supplementary Fig. 2 and Source data. Ub ubiquitin.

ubiquitin cannot reach the Ube2N/Ube2V2 binding surface (Fig. 3a). Consistent with this, we found that TRIM21 ubiquitin transfer in *trans* can occur in principle (Supplementary Fig. 7), in line with previous work on TRIM5 (ref. [20]).

To investigate the spatial requirements of TRIM21 RING domains for self-anchored ubiquitination experimentally, we established a substrate-dependent ubiquitination assay. TRIM21 is recruited by Fc, which is an obligate dimer in solution and can be bound by two PRYSPRY (PS) domains[30] (Supplementary Fig. 8). To test for the catalytic RING topology, we designed a series of mono-ubiquitinated TRIM21 constructs that vary the number of RINGs available and their distance to each other when bound to Fc (Fig. 3b and Supplementary Fig. 8). To suppress background activity, TRIM21 was used at low concentrations (100 or 50 nM) and the reaction was incubated for 5 min only. Full-length TRIM proteins form antiparallel homodimers via their coiled-coil domains, resulting in the separation of the two TRIM21 RING domains by ~17 nm even when bound to Fc (Supplementary Fig. 8). According to our model, addition of Fc alone should therefore not induce the catalytic RING topology (Fig. 3c). Indeed, addition of Fc did not stimulate ubiquitination of the full-length Ub-TRIM21 (Ub-RING-Box-coiled-coil-PRYSPRY or Ub-R-B-CC-PS, Fig. 3d). Even when adding an additional RING domain to make the full-length protein a constitutive RING dimer (Ub-R-R-B-CC-PS), formation of the catalytic RING topology is excluded (Fig. 3c) and no induction of self-ubiquitination is observed upon addition of Fc (Fig. 3d, and Supplementary Figs. 8 and 9). As a next step, we designed

TRIM21 constructs lacking the B-box and coiled-coil (Ub-R-PS and Ub-R-R-PS). Fc is capable of recruiting two of these constructs, thereby locating their RINGs within ~9 nm (Fig. 3c and Supplementary Fig. 8), the distance required for the catalytic RING topology (Fig. 3a, c). Addition of Fc to Ub-R-PS led to weak self-ubiquitination. This low level of activity is likely because Ub-R-PS can only provide a monomeric RING as the enzyme, while a monomeric RING on the second Ub-R-PS acts as the substrate. TRIM RING dimerization is known to greatly increase ligase activity[21,31–33]. We therefore repeated these experiments using a Ub-R-R-PS construct. We predicted that this should allow the catalytic RING topology observed in our crystal structure to form upon substrate binding, as the Fc will bring two RING dimers into close proximity (Fig. 3a, c). Indeed, addition of Fc to Ub-R-R-PS resulted in the efficient formation of TRIM21-anchored ubiquitin chains (Fig. 3d). Importantly, while anchored ubiquitination occurred very efficiently, hardly any free ubiquitin chains could be observed (Supplementary Fig. 9c). Since self-ubiquitination only requires E2~Ub to be recruited by the ligase, this explains its high efficiency relative to free ubiquitin chain formation, as the latter would require recruitment of both E2~Ub and (poly-) ubiquitin. Indeed, Ub-R-R-PS worked efficiently in our substrate-induced ubiquitination assay even at reduced TRIM21 concentrations (Supplementary Fig. 9d). Thus, inducing formation of the catalytic RING topology by substrate binding enables robust and selective formation of self-anchored ubiquitin chains. Moreover, the catalytic RING topology is only achieved when the separate requirements of an active enzyme (a

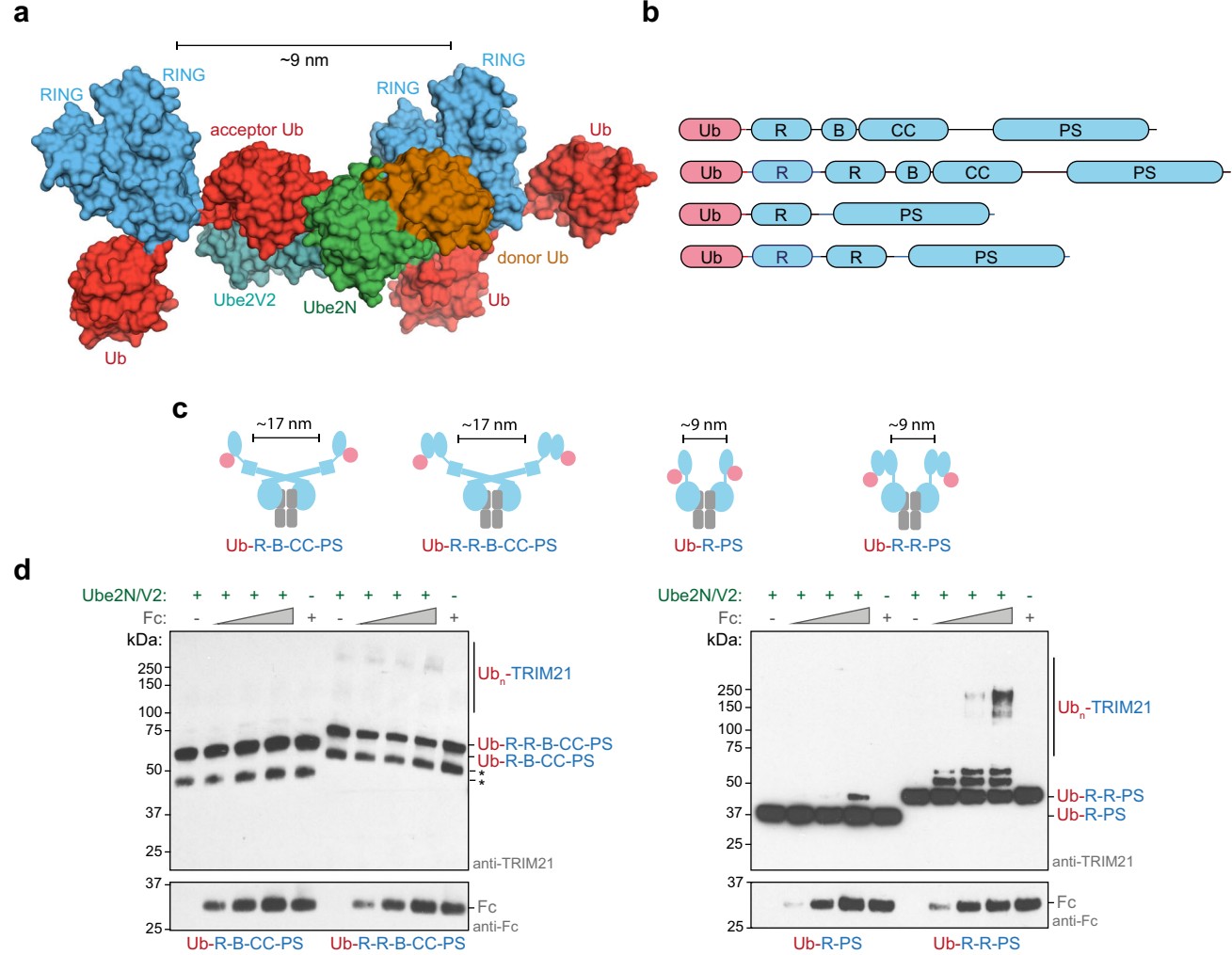

**Fig. 3 The mechanism of RING-anchored ubiquitination in *trans*. a** Surface representation of the canonical model of the Ub-R:Ube2N-Ub:
Ube2V2 structure (Ub-R, Ub in red, R in blue, Ube2N-Ub, Ube2N in green, Ub in orange, and Ube2V2 in teal). **b** Domain architecture of TRIM21 constructs
used in biochemical assays. **c** Cartoon models of substrate (Fc, gray) engagement by TRIM21 constructs (blue). The structural basis for these models is
shown in Supplementary Fig. 8. **d** Substrate (Fc)-induced self-ubiquitination assay of 100 nM Ub-TRIM21 constructs. Reactions were incubated for 5 min at
37 °C. Further data can be found in Supplementary Fig. 9. * (asterisk) indicates a TRIM21 degradation product that could not be removed during
purification. Western blots are representative of $n = 3$ independently performed experiments. Uncropped blots are provided in Source data. Ub ubiquitin, R
RING, B Box, CC coiled-coil, PS PRYSPRY, kDa kilo Dalton.

dimeric RING) and a correctly positioned substrate (a third
RING) are fulfilled.

We next considered how long a TRIM21-anchored ubiquitin
chain would have to be for *cis* ubiquitination to become sterically
possible. Using our Ub-R:Ube2N~Ub:Ube2V2 structure, we
created models with increasing numbers of K63-linked ubiquitin
chains conjugated to the TRIM21 RING domain. These models
suggested that a chain of four ubiquitin molecules would be
necessary and sufficient for self-ubiquitination in *cis* (Fig. 4a and
Supplementary Fig. 10). Thus, after addition of the priming
ubiquitin, three ubiquitin molecules must be added in *trans*,
before the chain could be further elongated in *cis*. Consistent with
this, we only observed very long TRIM21-anchored ubiquitin
chains or species carrying one, two, or three ubiquitin molecules
in our Fc-dependent TRIM21 ubiquitination experiments (Fig. 3d
and Supplementary Fig. 9c, d). With the addition of a fourth
ubiquitin, the reaction appears to progress much more quickly, as
would be expected for a switch from *trans* to *cis*, rapidly
consuming the tetra-ubiquitin species and converting it into a
long chain. In the above experiments, self-ubiquitination only

occurred when two Ub-R-R-PS constructs were colocalized by
their binding to Fc to satisfy the requirements of the catalytic
RING topology (Fig. 3c, d). To confirm the switch in self-
ubiquitination from *trans* to *cis* experimentally, we generated
TRIM21 R-R-PS constructs, wherein their N-termini were fused
to up to four linearly connected ubiquitin molecules. Due to their
high structural similarity[34], we assumed a linear chain would
mimic a K63-linked ubiquitin chain in length and flexibility
sufficiently well. Upon testing these new constructs, we observed
that only TRIM21 modified with tetra-ubiquitin became
independent of Fc for self-ubiquitination (Fig. 4b). All the other,
shorter, constructs remained rate-limited by first having to self-
ubiquitinate in *trans*, before switching to *cis*. This biochemical
data is in agreement with our structure, showing the initiation of
RING-anchored ubiquitination in *trans* and our model of
polyubiquitinated RING elongation in *cis*.

Finally, we considered whether the catalytic RING topology is
an arrangement specific to Ube2N or one that also works with
other E2 enzymes. Thus, we tested whether addition of Fc could
induce self-ubiquitination of Ub-TRIM21 in presence of Ube2D1,

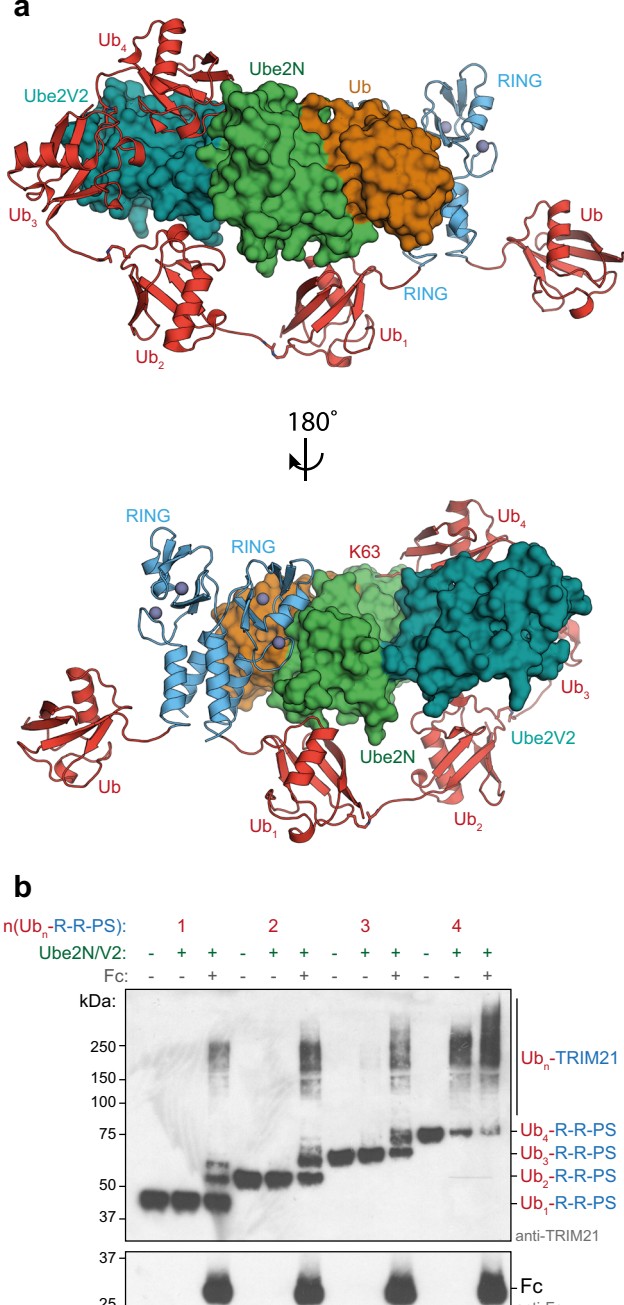

**Fig. 4 The mechanism of RING-anchored ubiquitination in *cis*. a** For ubiquitination in *cis*, the RING-anchored (blue) ubiquitin chain (red) must be sufficiently long to reach the active site on Ube2N~Ub/Ube2V2 (Ube2N in green, Ub in orange, and Ube2V2 in teal). The chain can go around two different routes, one shown here and the other in Supplementary Fig. 10. The ubiquitin chain was modeled using the Ub-R:Ube2N-Ub: Ube2V2 structure and a K63-linked $Ub_2$ structure (2JF5 (ref. [34])) using PyMol. **b** Substrate (Fc)-induced self-ubiquitination assay of 100 nM $Ub_n$-TRIM21 constructs. Reactions were incubated for 5 min at 37 °C. Western blots are representative of $n = 3$ independently performed experiments. Uncropped blots are provided in Source data. Ub ubiquitin, R RING, PS PRYSPRY, kDa kilo Dalton.

a highly promiscuous E2 enzyme. However, even after extended reaction times hardly any TRIM21 modification was detected, while in contrast free ubiquitin chains could be observed (Supplementary Fig. 11). The catalytic RING topology we observe

in our structure is thus specific for Ube2N/Ube2V2, explaining why this enzyme[16] and not Ube2D1 is required for TRIM21's cellular function[17]. Moreover, this may explain why TRIM21, and other TRIMs, such as TRIM5, build K63- and not K48-linked ubiquitin chains when first activated. Their mechanism of activation, induction of the catalytic RING topology, only results in formation of self-anchored K63 chains by using Ube2N/Ube2V2. Collectively, these data identify formation of a catalytic *trans* RING topology, as the driving force behind self-ubiquitination of TRIM21 with Ube2N/Ube2V2.

**Catalytic RING topology drives targeted protein degradation.** Having established the RING topology necessary for self-anchored ubiquitination in vitro, we next investigated if this same arrangement is required for TRIM21 activity in cells. We designed a similar series of TRIM21 constructs for cellular expression as above, which control for the number of RINGs available and their distance to each other when bound to Fc (Fig. 5a). We expressed these constructs in *TRIM21* knockout RPE-1 cells together with GFP-tagged Fc and monitored GFP-Fc degradation as a readout for TRIM21 activity, in a targeted protein degradation experiment. Consistent with the inability to form anchored chains when engaged with Fc in vitro, full-length TRIM21 did not degrade GFP-Fc in cells (Fig. 5b, c). Degradation could not be rescued by addition of another RING to the N-terminus, presumably because in this case the RINGs are dimeric but still separated by the coiled-coil, with the consequence that no "substrate" RING is available for ubiquitination. Thus, RING dimerization is not sufficient for cellular TRIM21 activity. In the R-PS construct, the RINGs are within ~9 nm, and thus within the range compatible with activity as defined by our structure (Fig. 3a). Despite this, no degradation was observed (Fig. 5b, c), likely because the RINGs can either form a single dimer, or one monomer RING would have to act as the enzyme and the other RING as the substrate. This is consistent with the inefficient self-ubiquitination of a comparable construct in our biochemical experiments (Fig. 3d). Only R-R-PS showed efficient GFP-Fc degradation (Fig. 5b, c). When this construct engages Fc, two RING dimers can form in close proximity, so that one RING dimer is available to mediate the ubiquitination of the other, thus fully satisfying the requirements of the catalytic RING topology. All the constructs were expressed at comparable levels and were active in classical Trim-Away targeted protein degradation assays (Fig. 5d, e and Supplementary Fig. 12), suggesting that the only difference is the number and relative distance of RING domains when engaged with the GFP-Fc construct. This also agrees with our biochemical data, where a similar construct shows strong self-ubiquitination upon substrate binding (Fig. 3d). Therefore, the Fc-induced self-ubiquitination assay in vitro provides a good prediction for the cellular activity. Our crystal structure of the initiation of RING-anchored ubiquitin chain elongation therefore precisely visualizes how this process can work in a physiological context.

## Discussion

Protein ubiquitination is one of the most abundant posttranslational modifications, affecting essentially all cellular events. A precise understanding of its underlying mechanisms, and how they achieve selectivity, is desirable for several reasons. First, malfunction in the ubiquitin system often manifests in severe disease[35]. Second, the use of small molecules (PROTACs, molecular glues, etc.) has emerged as a promising new approach for targeted degradation of disease-causing proteins in patients[36–41]. Despite recent advances in the field, it remains unclear how RING E3 ligases achieve specific substrate ubiquitination and how a

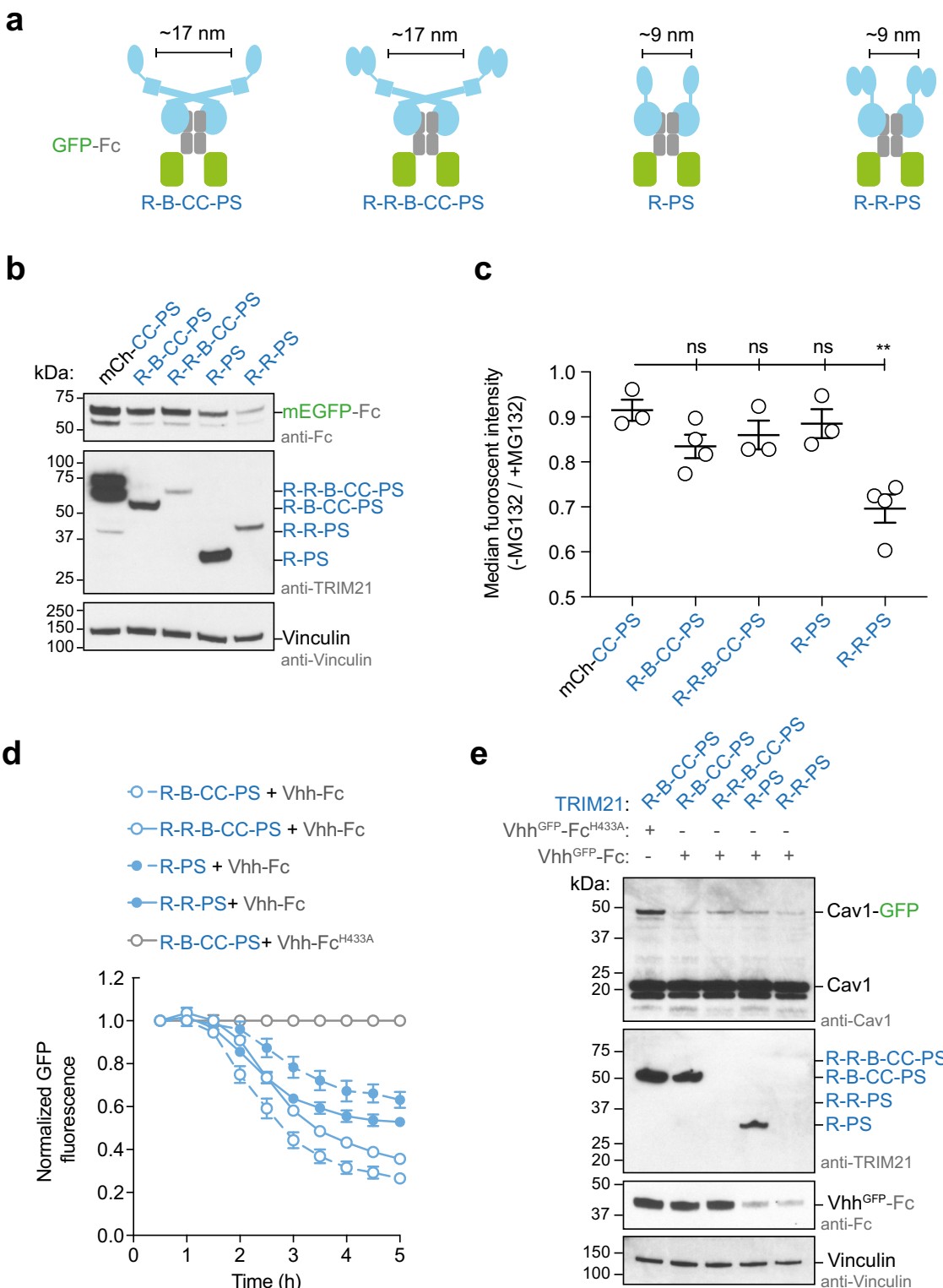

ubiquitin chain is formed after the priming ubiquitin has been transferred.

Here, we provide a structural framework for understanding RING E3-anchored ubiquitin chain formation. We were able to capture a snapshot of this process in a crystal structure of Ub-R with the ubiquitin-charged heterodimeric E2 enzyme Ube2-N~Ub/Ube2V2 (Fig. 1), showing the chemical activation of the acceptor ubiquitin, exemplified by the deprotonation of the acceptor lysine by Ube2N D119 (Fig. 2). Most importantly, our

structure reveals the domain arrangement required for the elongation reaction, in other words a catalytic RING E3 topology that enables the extension of a mono-ubiquitinated RING into a K63-linked, RING-anchored ubiquitin chain (Figs. 3 and 4). In this arrangement, two RINGs form a dimer and act as an enzyme on a third RING domain, which acts as the substrate in this reaction. We observe that while rigidity is required to position all the important catalytic residues in the E2 active site optimally (Fig. 2 and Supplementary Fig. 1), formation of the substrate-anchored

**Fig. 5 Catalytic RING topology drives targeted protein degradation. a** Schematic cartoons showing the topology of TRIM21 (blue) on GFP-Fc (green and gray, respectively). **b**, **c** GFP-Fc degradation assay. **b** Western blot of RPE-1 *TRIM21*-knockout cells transiently expression GFP-Fc and a series of TRIM21 constructs. Western blots are representative of $n = 2$ independently performed experiments. **c** Shown is the flow cytometry analysis of green fluorescence of RPE-1 *TRIM21*-knockout cells transiently expressing GFP-Fc and a series of TRIM21 constructs. After electroporation, each population of cells was split in two and either treated with MG132 or DMSO. Data are presented as mean ± standard error of the mean. Each data point in the graph represents one biologically independently performed experiment ($n = 3$ (for mCh-CC-PS, R-R-B-CC-PS, and R-PS) or 4 (R-B-C-C-PS and R-R-PS)). A two-tailed unpaired Student's *T* test was performed to assess the significance of fluorescence reduction relative to mCh-CC-PRYSPRY (*P* values: R-B-CC-PRYSPRY, 0.0797 (ns); R-R-B-CC-PRYSPRY, 0.02366 (ns); R-PRYSPRY, 0.4964 (ns); and R-R-PRYSPRY, 0.0035 (**)). **d**, **e** Trim-Away of Caveolin-1-mEGFP (Cav1-GFP) in NIH 3T3 *GFP-Cav-1-knock in* cells[66]. Shown in **d** is the normalized GFP fluorescence (error bars represent ± SEM of four images) and in **e** the western blot after the experiment. Data in **d**, **e** are representative of $n = 2$ independent experiments. Uncropped blots and raw data are provided in Source data. R RING, B Box, CC coiled-coil, PS PRYSPRY, mCh mCherry, kDa kilo Dalton, ns not significant.

ubiquitin chain likely requires conformational flexibility between domains that is provided by the unique topology of TRIM proteins (Fig. 3). Substrate-induced self-ubiquitination of TRIM21 is highly efficient, even at low ligase concentration, in contrast to free ubiquitin chain formation (Fig. 3 and Supplementary Fig. 9). This implies that physiological ubiquitin signals may not be produced as free chains but mainly on substrates, due to the higher reaction efficiency.

Our data establishes that the RING-anchored K63-chain is first formed in a *trans* mechanism, where a RING dimer activates a Ube2N~Ub molecule, thereby acting as an E3 ligase. An additional mono-ubiquitinated RING acts as a substrate for ubiquitination and accepts the donor ubiquitin (Fig. 3). Only after four ubiquitin molecules have been added to the RING in *trans*, is the chain sufficiently long for ubiquitin chain formation in *cis* (Fig. 4). While ubiquitin chain elongation in *cis* occurs at much higher rates, the initial need for a *trans* arrangement may represent an important regulatory mechanism suppressing TRIM21 activity in absence of a substrate. In the case of TRIM21 or TRIM5α, activation is driven by substrate binding, which is needed for *trans* ubiquitination. Interestingly, substrate modification with linear ubiquitin chains by the RBR (RING-in between-RING) ligase HOIP is regulated by its partner RBR HOIL, which mono-ubiquitinates all three LUBAC components HOIP, HOIL, and SHARPIN. These ubiquitin primers are then elongated in *cis* by HOIP, thereby outcompeting *trans* ubiquitination of substrates[42]. Thus, switching between *cis* and *trans* mechanisms of ubiquitination may be a regulatory system exploited by many different types of E3 ligases.

The catalytic RING topology observed in our structure predicts the requirements for TRIM21-mediated targeted protein degradation in cells (Fig. 5). Upon substrate recognition, TRIM21 forms a K63-linked ubiquitin chain on its N-terminus[13]. Loss of this K63-linked ubiquitin chain prevents virus neutralization, immune signaling, and Trim-Away[17]. Our GFP-Fc degradation experiment shows that only the TRIM21 construct (R-R-PS) that can form the catalytic RING topology under these conditions enables degradation (Fig. 5). Interestingly, specific orientation of the E3 ligase CRL^VHL relative to its substrate was also shown to be critical for targeted protein degradation[43]. Nevertheless, how a RING-anchored K63-chain leads to degradation of the RING and its bound substrate remains mysterious. A mechanism could be envisioned, whereby this chain provides the scaffold for a branching event that allows the synthesis of degradation competent K48-linked ubiquitin chains. Indeed, TRIM21 can be modified with both K63- and K48-linked ubiquitin chains and K48 chain formation is dependent upon, and occurs subsequent to, K63-ubiquitination[13]. Degradation of the proapoptotic regulator TXNIP for instance was shown to be mediated by K48-chains that were assembled on substrate-bound K63 chains[44]. K63–K48-chains have also been shown to amplify NF-κB

signals[45]. A strategy of mixed, branched chains might be essential for TRIM21 to act as both an immune sensor and effector[13].

The catalytic RING topology we describe is consistent with data showing that TRIM proteins can undergo higher-order assembly. In the case of TRIM5α[46], three TRIM5α RINGs are brought into close proximity when the protein is incubated with the HIV capsid[46–48] (Fig. 6a, b and Supplementary Fig. 13). This positioning would fulfill the catalytic RING topology we describe, and would be consistent with the ability of TRIM5α to restrict retroviruses[49,50] and activate the innate immune response via self-anchored K63-ubiquitination[12,51]. The functional requirement for multiple TRIM molecules is also suggested by the fact that potent antibody-mediated neutralization of adenovirus by TRIM21 requires multiple antibodies bound per virus[52]. In addition, TRIM21 was shown to be activated by substrate-induced clustering, resulting in multiple TRIM21:antibody complexes on the substrate[53]. The unique TRIM architecture, in which the RINGs are located at either end of a coiled-coil, and the flexibility provided by the hinge region of the antibody, may be crucial in enabling TRIM21 molecules bound onto the surface of a virus to engage with each other (Fig. 6c). To fulfill the catalytic RING topology on the virus, two RINGs need to dimerize and a third has to be within ~9 nm of the RING dimer, enabling self-anchored ubiquitination and subsequent virus neutralization (Fig. 6d). Since higher-order assembly has been associated with many other K63 ubiquitin chain forming RING E3 ligases, such as TRAF6 (ref. [54]), RIPLET[55], and others, we propose that the mechanism presented here is thus likely to be found more widely within the realm of RING E3 ligases.

## Methods

**Plasmids.** Bacterial expression constructs: Ube2V2 and TRIM21 expression constructs, but full-length were cloned into pOP-TG vectors and full-length TRIM21 constructs into HLTV vectors. Ube2N constructs were cloned into pOP-TS, Ube1 into pET21, and ubiquitin into pET17b. Ube2D1 was cloned into pET28a. For cloning Ub$_{4/3/2}$-TRIM21 constructs, a linear Ub$_3$ sequence was codon optimized, ordered as synthetic DNA (Integrated DNA technologies, Coralville, Iowa, USA) and inserted into the Ub^G75/76A-TRIM21 construct. All constructs for mRNA production were cloned into pGEMHE vectors[56]. Constructs were cloned by Gibson Assembly and mutations were inserted by mutagenesis PCR. For mCherry-TRIM21^ΔRING-Box, TRIM21^382–1428 was amplified by PCR and cut by EcoRI and NotI. A 743 bp fragment carrying mCherry was cut by AgeI and EcoRI from V60 (pmCherry-C1, Clonetech) and both fragments were ligated into pGEMHE. Primers, a complete list of all plasmids used in this study, and the primary structure of all purified proteins and mRNA that was expressed in cells are given in Supplementary Data 1.

**Expression and purification of recombinant proteins.** Ubiquitin-TRIM21, TRIM21 RING (residues 1–85), Ube2N, and Ube2V2 constructs were expressed in *Escherichia coli* BL21 DE3 cells. Ubiquitin and Ube1 were expressed in *E. coli* Rosetta 2 DE3 cells. All cells were grown in 2×TY media supplemented with 2 mM MgSO$_4$, 0.5% glucose and 100 μg mL$^{-1}$ ampicillin (and 35 μg mL$^{-1}$ chloramphenicol for expression is Rosetta 2 cells). Cells were induced at an OD$^{600}$ of 0.7. For TRIM proteins, induction was performed with 0.5 mM IPTG and 10 μM ZnCl$_2$, for ubiquitin and Ube1 with 0.2 mM IPTG and for E2 enzymes with 0.5 mM IPTG. After centrifugation, cells were resuspended in 50 mM Tris pH 8.0,

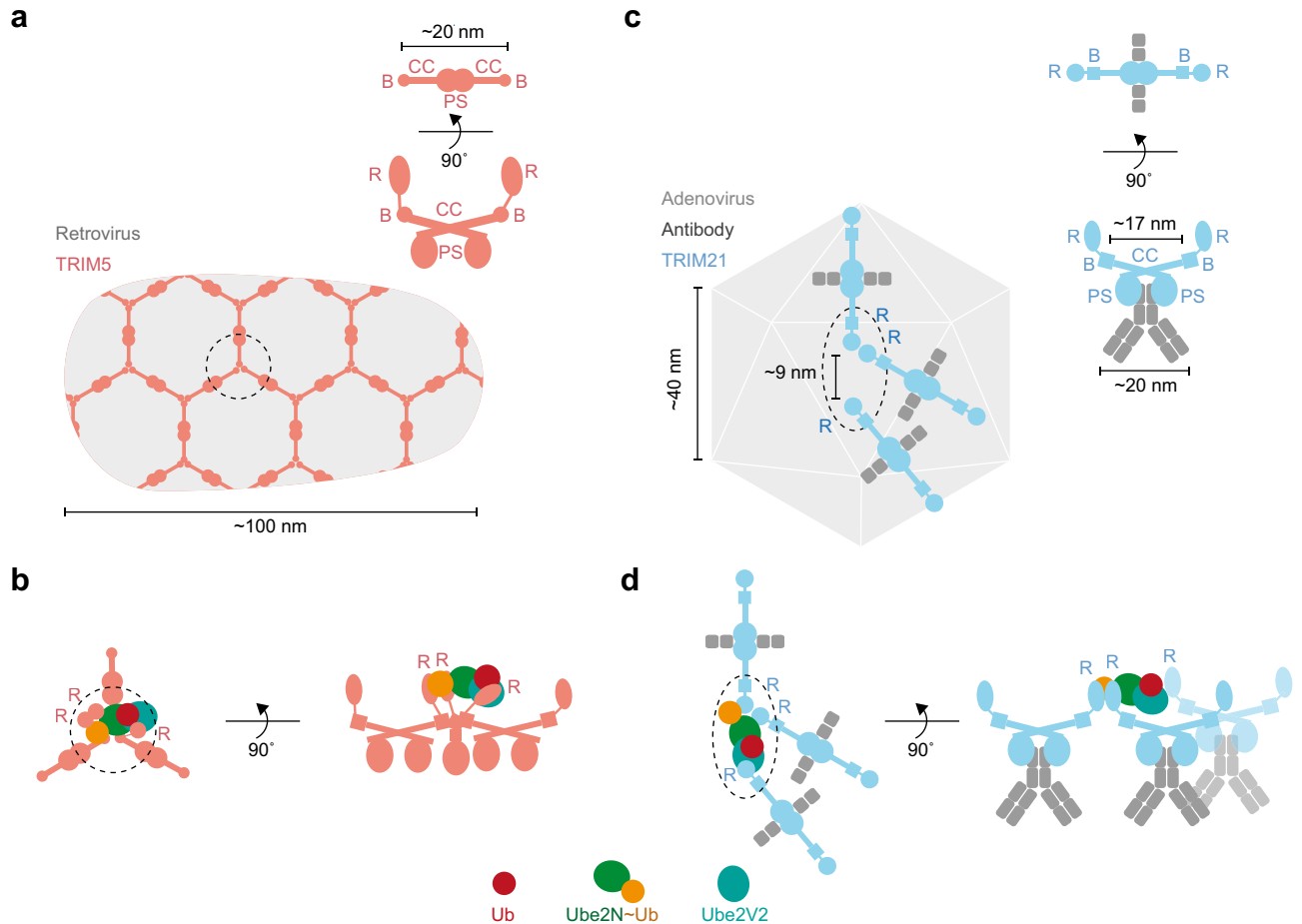

**Fig. 6 TRIM protein assembly on viruses.** Cartoon models of the assembly of TRIM5 (**a**, **b**) and TRIM21 (**c**, **d**) on viral capsids. **a** Shown is the hexagonal assembly of TRIM5 on HIV-1 capsid as imaged by cryo-electron tomography[48]. **c** Assembly of TRIM21:antibody complexes on adenovirus capsid (adenoviral measurements are based on 6B1T[68]). **b**, **d** Cartoons visualizing how the TRIM protein assembly on the viral capsid enables the formation of the catalytic RING topology. R RING, B Box, CC coiled-coil, PS PRYSPRY, Ub ubiquitin.

150 mM NaCl, 10 μM ZnCl₂, 1 mM DTT, 20% Bugbuster (Novagen), and cOmplete protease inhibitors (Roche, Switzerland). Lysis was performed by sonication. TRIM proteins and Ube2V2 were expressed with N-terminal GST-tag and purified via glutathione sepharose resin (GE Healthcare) equilibrated in 50 mM Tris pH 8.0, 150 mM NaCl, and 1 mM DTT. The tag was cleaved on beads overnight at 4 °C. In case of ubiquitin-TRIM21 constructs, the eluate was supplemented with 10 mM imidazole and run over 0.25 mL of Ni-NTA beads to remove His-tagged TEV. Ube2N and Ube1 were expressed with an N-terminal His-tag and were purified via Ni-NTA resin. Proteins were eluted in 50 mM Tris pH 8.0, 150 mM NaCl, 1 mM DTT, and 300 mM imidazole. For Ube2N, TEV-cleavage of the His-tag was performed overnight by dialyzing the sample against 50 mM Tris pH 8.0, 150 mM NaCl, 1 mM DTT, and 20 mM imidazole. Afterward, His-tagged TEV protease was removed by Ni-NTA resin. The cleavage left an N-terminal tripeptide scar (GSH) on recombinantly expressed TRIM proteins, an N-terminal G scar on Ube2N and an N-terminal GSQEF scar on Ube2V2. Finally, size-exclusion chromatography was carried out on a HiLoad 26/60 Superdex 75 prep grade column (GE Healthcare) in 20 mM Tris pH 8.0, 150 mM NaCl, and 1 mM DTT.

Full-length TRIM21 (Ub-R-B-CC-PS or Ub-R-B-CC-PS) were expressed as His-Lipoyl-fusion proteins in *E. coli* BL21 DE3 cells. Cells in 2×TY were grown to an OD600 of 0.8 and induced with 0.5 mM IPTG and 10 μM ZnCl₂. Cells were further incubated at 18 °C, 220 r.p.m. overnight. After centrifugation, cells were resuspended in 100 mM Tris pH 8.0, 250 mM NaCl, 10 μM ZnCl₂, 1 mM DTT, 20% Bugbuster (Novagen), 20 mM imidazole, and cOmplete protease inhibitors (Roche, Switzerland). Lysis was performed by sonication. His-affinity purification was performed as described above. Immediately afterward, the protein was applied to an S200 26/60 column (equilibrated in 50 mM Tris pH 8.0, 200 mM NaCl, and 1 mM DTT) to remove soluble aggregates. After concentration determination, the His-Lipyol tag was cleaved using TEV protease overnight. Since full-length TRIM21 is unstable without tag, the protein was not further purified but used for assays.

Ubiquitin purification was performed following the protocol established by the Pickart lab[57]. After cell lysis by sonication (lysis buffer: 50 mM Tris pH 7.4, 1 mg mL⁻¹ Lysozyme (by Sigma Aldrich, St. Louis, USA) and 0.1 mg mL⁻¹ DNAse (by Sigma

Aldrich, St. Louis, USA)), a total concentration of 0.5% percloric adic was added to the stirring lysate at 4 °C. The (milky) lysate was incubated for another 30 min on a stirrer at 4 °C to complete precipitation. Next, the lysate was centrifuged (50,000 × g) for 30 min at 4 °C. The supernatant was dialyzed overnight (3500 MWCO) against 3 L 50 mM sodium acetate (NaOAc) pH 4.5. Afterward, Ub was purified via cation exchange chromatography using a 20 mL SP column (GE Healthcare) and using a NaCl gradient (0–1000 mM NaCl in 50 mM NaOAc pH 4.5). Finally, size-exclusion chromatography was carried out on a HiLoad 26/60 Superdex 75 prep grade column (GE Healthcare) in 20 mM Tris pH 7.4.

All proteins were flash frozen in small aliquots (30–100 μL) and stored at −80 °C.

**Formation of an isopeptide-linked Ube2N-Ub.** Ube2N^C87K/K92A charging with WT ubiquitin was performed as normal E1-mediated charging, but in a high pH to ensure K87 deprotonation[10,17]. The isopeptide charging reaction was carried out in 50 mM Tris pH 10.0, 150 mM NaCl, 5 mM MgCl₂, 0.5 mM TCEP, 3 mM ATP, 0.8 μM Ube1, 100 μM Ube2N, and 130 μM ubiquitin at 37 °C for 4 h. After conjugation, Ube2N^C87K/K92A~Ub was purified by size-exclusion chromatography (Superdex S75 26/60, GE Healthcare) that was equilibrated in 20 mM Tris pH 8.0 and 150 mM NaCl.

**Crystallization.** In total, 5 mg mL⁻¹ of human Ub^G75/76A-TRIM21^1–85, Ube2N^C87K/K92A~Ub, and Ube2V2 in 20 mM Tris pH 8.0, 150 mM NaCl, and 1 mM DTT were subjected to sparse matric screening in sitting drops at 17 °C (500 nL protein was mixed with 500 nL reservoir solution). Crystals were obtained in Morpheus II screen[58] in 0.1 M MOPSO/bis-tris pH 6.5, 12.5% (w/v) PEG 4 K, 20% (v/v) 1,2,6-hexanetriol, and 0.03 M of each Li, Na, and K.

For the Ube2N^C87K/K92A~Ub:Ube2V2 structure, 10 mg mL⁻¹ TRIM21^1–85, Ube2N^C87K/K92A~Ub, Ube2V2, and Ub in 20 mM Tris pH 8.0, 150 mM NaCl, and 1 mM DTT were subjected to sparse matrix screening in sitting drops at 17 °C (200 nL protein was mixed with 200 nL reservoir solution). Crystals were obtained in the Morpheus III screen[59] in 0.1 M bicine/Trizma base pH 8.5, 12.5% (w/v) PEG 1000, 12.5% (w/v) PEG 3350, 12.5% (v/v) MPD, and 0.2% (w/v) of each anesthetic

alkaloids (lidocaine HCl · $H_2O$, procaine HCl, proparacaine HCl, and tetracaine HCl). Crystals were flash frozen for data collection without the use of additional cryo-protectant.

**Crystal data collection, structure solution, and refinement**. Data were collected at the Diamond Light Source beamline i03, equipped with an Eiger2 XE 16 M detector of a wavelength of 0.9762 Å. For $Ub^{G75/76A}$-$TRIM21^{:1–85}Ube2N^{C87K/K92A}$~Ub: Ube2V2, diffraction images were processed using XDS[60] to 2.2 Å resolution. The crystals belong to space group number 5 (*C2*) with each of the components present as a single copy in the asymmetric unit. Analysis of the raw data revealed moderate anisotropy in the data. The structure was solved by molecular replacement using PHASER-MR implemented in the Phenix suite[61]. Search models served TRIM21 RING and Ube2N from 6S53 (ref. [17]), ubiquitin from 1UBQ[62] and Ube2V2 from 1J74 (ref. [7]). Model building and real-space refinement was carried out in coot[63], and refinement was performed using phenix-refine[64]. The anisotropy in the data could be observed in parts of the map that were less well resolved. While all interfaces show clear high-resolution density, particularly parts of Ube2V2 (chain A) that were next to a solvent channel proved challenging to build. The structure is deposited in the Protein Data Bank under the accession code 7BBD.

For $Ube2N^{C87K/K92A}$~Ub:Ube2V2, diffraction images were processed using XDS[60] to 2.54 Å resolution. The crystals belong to space group number 145 ($P3_2$) with each component present three times in the asymmetric unit, related by translational non-crystallographic symmetry. The structure was solved by PHASER-MR implemented in the Phenix suite[61]. Search models used were Ube2N from 6S53 (ref. [17]), ubiquitin from 1UBQ[62], and Ube2V2 from 1J74 (ref. [7]). Model building and real-space refinement was carried out in coot[63], and refinement was performed using phenix-refine[64]. The structure is deposited in the Protein Data Bank under the accession code 7BBF.

**Ubiquitin chain formation assay**. Ubiquitin chain formation assays were performed in 50 mM Tris pH 7.4, 150 mM NaCl, 2.5 mM $MgCl_2$, and 0.5 mM DTT. The reaction components were 2 mM ATP, 0.25 μM Ube1, 80 μM ubiquitin, 0.5 μM Ube2N/Ube2V2, or Ube2D1 together with the indicated concentration of E3. Samples were taken at the time points indicated and the reaction was stopped by addition of LDS sample buffer at 4 °C. The samples were boiled at 90 °C for 2 min and resolved by LDS–PAGE. Ubiquitin chains were detected in the western blot using an anti-Ub-HRP (Santa Cruz, sc8017-HRP P4D1, 1:5000), TRIM21 by rabbit anti-TRIM21$^{PRYSPRY}$ D101D ST#9204 (1:1000), and Fc by goat anti-human IgG-Fc broad 5211–8004 (1:2000).

**Kinetics of di-ubiquitin formation**. Kinetic measurements of di-ubiquitin formation were measured for Michaelis–Menten and $pK_a$ analysis. The experiment was performed in a pulse-chase format, where the first reaction generated Ube2N~$^{His}$-Ub and was chased by $Ub^{1-74}$. Under these conditions, $Ub^{1-74}$ only acts as acceptor, as it cannot be charged onto the E1 enzyme. His-tagged ubiquitin on the other hand serves as donor. Although, theoretically His-Ub could also act as an acceptor, the high concentrations of $Ub^{1-74}$ outcompete His-Ub as an acceptor. Initially, we determined the linear phase of the reaction for all different constructs, so as to later measure only one point on this trajectory as a representative for the initial velocity ($v_0$). For Michaelis–Menten kinetics, we used the following length: WT, 3 min; D119A, 100 min; D119N, 30 min; N123A, 3 min; D124A. 3 min, and for $pK_a$ measurements the following: WT, 40 s; D119A, 5 min; D119N, 60 s; N123A, 40 s; D124A, 40 s.

First, Ube2N charging was performed in 50 mM Tris pH 7.0, 150 mM NaCl, 20 mM $MgCl_2$, 3 mM ATP, 60 μM His-ubiquitin, 1 μM GST-Ube1 (Boston Biochem), and 40 μM Ube2N. The reaction was incubated at 37 °C for 12 min and stored afterward at 4 °C until use (within 1 h).

For Michaelis–Menten kinetic analysis, the reaction was conducted in 50 mM Tris pH 7.4, 150 mM NaCl with the indicated amount of $Ub^{1-74}$ (0–400 μM), while for $pK_a$ determination in 50 mM Tris and the indicated pH (7.0–10.5), 50 mM NaCl, and 250 mM $Ub^{1-74}$. Apart from the buffer, the reaction mix contained 2.5 μM Ube2V2. The reaction was initiated by addition of charging mix that was diluted 1 in 20, resulting in 2 μM Ube2N in the reaction. The reaction was stopped by addition of 4× LDS loading buffer. The samples were boiled at 90 °C for 2 min and resolved by LDS–PAGE. Western blot was performed with anti-His antibody (Clontech, 631212, 1:5000) via the LiCor system, leading to detection of the following species: His-Ub, His-Ub-$Ub^{1-74}$, Ube2N~$^{His}$-Ub, Ube2N~($^{His}$Ub)$_2$ (a side product of the charging reaction that shows ubiquitination rates similar to Ube2N~$^{His}$-Ub), and E1-$^{His}$-Ub. The concentration of $^{His}$-Ub-$Ub^{1-74}$ was determined by dividing the value for $^{His}$-Ub-$Ub^{1-74}$ by the sum of all bands detected and multiplying this by the total concentration of $^{His}$-Ub in the reaction (3 μM). Experiments were performed in technical triplicates. Michaelis–Menten kinetics data were fit to Eq. (1):

$$V = \frac{E_t * k_{cat} * S}{K_M * S} \tag{1}$$

where $V$ is the measured velocity, $E_t$ the total concentration of active sites (2 μM), and $S$ the substrate concentration. The curve was fit to determine $k_{cat}$ and $K_M$. To determine the $pK_a$, the data was fit to Eq. (2):

$$V = \frac{V_{HA} * 10^{-pH} + V_{A^-} * 10^{-pK_a}}{10^{-pK_a} + 10^{-pH}} \tag{2}$$

as given in ref. [65], where $V$ is the measured velocity, $V_{A^-}$ the velocity for the basic species, and $V_{HA}$ the velocity for the acidic species.

**In vitro transcription and RNA purification**. For in vitro transcription of mRNA, constructs were cloned into pGEMHE vectors[56]. Plasmids were linearized, using AscI. Capped (but not polyA-tailed) mRNA was synthesized with T7 polymerase, using the HiScribe™ T7 ARCA mRNA Kit (New England Biolabs) according to the manufacturer's instructions.

**Cell lines**. NIH3T3-Caveolin-1-EGFP[66] were cultured in DMEM medium (Gibco; 31966021) supplemented with 10% calf serum and penicillin–streptomycin. RPE-1 cells (ATCC) were cultured in DMEM/F-12 medium (Gibco; 10565018) supplemented with 10% calf serum and penicillin–streptomycin.

All cells were grown at 37 °C in a 5% $CO_2$ humidified atmosphere and regularly checked to be mycoplasma free. The sex of NIH3T3 cells is male. The sex of RPE-1 cells is female. Following electroporation, cells were grown in medium supplemented with 10% calf serum without antibiotics. For live imaging with the IncuCyte (Sartorius), cell culture medium was replaced with Fluorobrite (Gibco; A1896701) supplemented with 10% calf serum and GlutaMAX (Gibco; 35050061).

RPE-1 TRIM21 knockout cells were generated using the Alt-R CRISPR-Cas9 system from Integrated DNA technologies (IDT) with a custom-designed crRNA sequence (ATGCTCACAGGCTCCACGAA). Guide RNA in the form of crRNA-tracrRNA duplex was assembled with recombinant Cas9 protein (IDT #1081060) and electroporated into RPE-1 cells together with Alt-R Cas9 Electroporation Enhancer (IDT #1075915). Two days post electroporation, cells were plated one cell per well in 96-well plates and single-cell clones screened by western blotting for TRIM21 protein. A single clone was chosen that contained no detectable TRIM21 protein and confirmed TRIM21 knockout phenotype in a Trim-Away assay.

For the proteasome inhibition experiments, MG132 (Sigma; C2211) was used at a final concentration of 25 μM.

**Transient protein expression from mRNA**. To enable precise control of protein expression levels, constructs were expressed from in vitro transcribed mRNA. mRNA was delivered into cells by electroporation using the Neon Transfection system (Invitrogen). For each electroporation reaction, $8 \times 10^5$ RPE-1 TRIM21-knockout or NIH3T3-Caveolin-1-EGFP cells suspended in 10.5 μl of resuspension buffer R were mixed with 2 μL of the indicated mRNA in water. After electroporation, cells were transferred into antibiotic-free DMEM or DMEM/F-12 media supplemented with 10% FBS and left to incubate for 5 h before cells were harvested. Typically, expression could be detected from 30 min after electroporation and lasted for ~24 h.

**Trim-Away**. For each electroporation reaction, $8 \times 10^5$ NIH 3T3 Cav1-knock in cells[66] suspended in 10.5 μl of resuspension buffer R were mixed with the indicated amount of antibody mixture diluted in 2 μL of PBS. mRNAs were added immediately prior to electroporation, to limit the degradation by potential RNAse activity. Cav1-GFP mRNA encoding Vhh-Fc (WT or PRYSPRY binding-deficient H433A mutant) and TRIM21 were electroporated. The cell mRNA mixtures were taken up into 10 μL Neon electroporation pipette tips (Invitrogen) and electroporated using the following settings: 1400 V, 20 ms, 2 pulses (as described in refs. [18,67]). Electroporated cells were transferred to antibiotic-free Fluorobright media supplemented with 10% FBS and left to incubate for 5 h in an incubator before the cells were harvested for immunoblotting. GFP fluorescence measured using an Incucyte® (essenbioscience) and was normalized to the control (Vhh-Fc$^{H433A}$). Protein detection was performed using the following antibodies: Fc: goat anti-hIgG Fc broad 5211–8004 (1:2000); TRIM21: rabbit anti-TRIM21 D101D (ST#9204) (1:1000), Vinculin: rabbit anti-Vinculin EPR8185 ab 217171 (1:50,000); and Caveolin-1: rabbit anti-Cav1 (BD: 610059, 1:1000).

**mEGFP-Fc degradation assay**. For mEGFP-Fc degradation assay, 0.4 μM mEGFP-Fc mRNA together with 1.2 μM of the indicated TRIM21 mRNA were electroporated into $8 \times 10^5$ cells, as described above. Electroporated cells were transferred to antibiotic-free DMEM supplemented with 10% FBS. For western analysis only, cells were incubated for 5 h in an incubator before harvest. For flow cytometry analysis, the half of the cells were taken and treated with 25 μM MG132, while the other half were treated with DMSO. Then cells were incubated for 5 h in an incubator before being harvested. Cells were fixed before being subjected to flow cytometry. The same antibodies were used as for Trim-Away (see above).

**Flow cytometry**. Cells were fixed prior to flow cytometry. For this, cells were resuspended in FACS fixative (4% formaldehyde, 2 mM EDTA in PBS) and incubated at room temperature for 30 min. Afterward, cells were centrifuged and resuspended in FACS buffer (2% FBS, 5 mM EDTA in PSB) and stored at 4 °C,

wrapped in aluminum foil until use. Flow cytometry was performed using an Eclipse (iCyt) A02-0058. Cells were measured using forward and side scattering to assess live cells. In addition, green fluorescence was measured. Live cells were selected based on forward and side scattering and only the median GFP fluorescence of live cells was used for further analysis (example data shown in Supplementary Fig. 12d).

**Software**. Figures were created using the following software: Graphpad Prism 7.0d, Pymol 1.8.2.3, Adobe Illustrator v24.2.3, Adobe Photoshop CS6 13.0.6 ×64, Image Studio Lite 5.2.5, and ImageJ/FIJI 2.0.0-rc-69/1.52p. For flow cytometry, we used ec800 v1.3.6, which is the operating software for the Eclipse (iCyt) A02-0058. For cell imaging using the Incucyte, we used the default software Incucyte S3. For crystallography, we used XDS (Version 31. January 2019), Phenix 1.18.2_3874 and 1.14-3260 (in these versions we used the implemented programs: Phaser, Phenix_Refine), and Coot 0.8.9 and 0.9.

**Reporting summary**. Further information on research design is available in the Nature Research Reporting Summary linked to this article.

## Data availability

The crystal structures of Ub-R:Ube2N~Ub:Ube2V2 and Ube2N~Ub:Ube2V2 are deposited in the Protein Data Bank under the accession codes 7BBD and 7BBF, respectively. All other relevant data are available from the corresponding authors upon reasonable request. Source data are provided with this paper.

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

## Acknowledgements

We would like to thank Dr. Marios G. Koliopoulos, Prof. Dr. Randy Read, Prof. Dr. Andreas Boland, Dr. Kyle W. Muir, Pablo Rodriguez-Silvestre, Laura E. Easton, Dr. Donna L. Mallery, Dr. Jane Wagstaff, Dr. Stefan MV Freund, and Dr. Viswanathan Chandrasekaran for discussion and/or technical aid. We would also like to thank Diamond light source (proposals mx15916-88 and mx21426-4) and the beamline staff for assistance (Dr. Pierre Aller and Mark Williams) and the LMB flow cytometry facility (Dr. Maria Daly). L.K. was supported by a Ph.D. Fellowship from the Boehringer Ingelheim Fonds. This work was supported by the MRC (UK, U105181010 to L.C.J. and U105178934 to D.N.) and a Wellcome Trust Investigator Award to L.C.J.

## Author contributions

Conceptualization, L.K. and L.C.J.; methodology, L.K., D.C., and L.C.J.; formal analysis; L.K.; investigation, L.K., D.C., and N.R.; writing—original draft, L.K.; writing—review and editing, L.K., D.C., N.R., D.N., and L.C.J.; supervision, D.N. and L.C.J.; and funding acquisition, L.K., D.N., and L.C.J.

## Competing interests

The authors declare no competing interests.
