## [Peer Review File · Nature Communications]

REVIEWER COMMENTS

Reviewer #1 (Remarks to the Author):

In their latest work, Kiss et al. report a crystal structure that captures TRIM21 in the act of building a polyubiquitin chain onto its N-terminus and define this as the catalytic arrangement that is required for signaling in response to viral detection. From a structural standpoint, this work builds upon previous work that has described individual elements such as E2 activation and polyubiquitin chain formation, but is the first to visualize the act of ubiquitin chain extension onto a “primed” monoubiquitinated substrate. From their structure, the authors observe interactions in the E2 catalytic center that activate the incoming substrate lysine. This interaction has previously been tested in other systems of ubiquitin conjugation, but the authors do a nice job of demonstrating its importance for chain assembly in their system. The bulk of the remaining work focuses on formation of the so-called “catalytic arrangement”, which sets strict distance parameters that are required for extension of the first few ubiquitin molecules in trans, after which the chain can be rapidly extended through additions in cis.

The work is of high quality and is described very clearly. As the authors point out, the experiments addressing activation of the incoming substrate lysine confirm a body of previous work that propose this mechanism. Perhaps the most striking result is the mechanism underlying formation of the catalytic arrangement. This model could be explained more clearly with additional discussion in the text and perhaps a cartoons schematic for a final figure, but the underlying principle is that the initial extension of self-anchored ubiquitin chains can only occur in trans if TRIM21 molecules are assembled onto a viral particle. Once the catalytic arrangement can be formed through this virus-induced oligomerization, ubiquitin chain extension can begin in trans and, after a certain threshold, can rapidly continue in cis to produce a K63-linked ubiquitin signal. This represents a new and interesting mechanism of ubiquitin signaling regulation. With some revisions to strengthen the communication of this model, I recommend publication.

Major comments:

1) Though the kinetic experiments presented in Figure 2 and Supplementary Figure 2 are well thought-out and internally consistent, there are several points that should be addressed to justify certain aspects of the methodology:

a. Is there any reason to believe that the auto-ubiquitinated form of Ube2N would not display altered rates of ubiquitin chain assembly? Is this accounted for in some way?

b. The His-Ub is presented as a dedicated donor ubiquitin and the Ub Δ GG as a dedicated acceptor, but without incorporating a K63R mutation into the His-Ub then I don't see how it can truly only act as a donor. I recognize that the concentration of His-Ub is much lower than Ub- Δ GG in the final chain assembly reaction, but formally it should still be possible for His-Ub to act as an acceptor. In fact, for

some of the slower reactions (e.g. Supplementary Figure 2C, D119A), a higher molecular weight di-ubiquitin band is visible that could reflect assembly of two His-Ub molecules. I don't think this could be affecting the results but it should be discussed.

c. For each panel in Supplementary Figure 2, the reaction timepoint selected for analysis should be listed.

2) As the more innovative product of this work, the mechanism of viral-scaffolded oligomerization that leads to the catalytic arrangement for chain extension deserves more focus.

a. The data presented in Figures 4 and 5 are compelling and nicely speak to your model for a catalytic arrangement requirement. Until I read about TRIM21 oligomerization on a viral scaffold in the discussion, however, I was left wondering what the biological relevance of your structure was if Fc-bound full-length TRIM21 could not activate chain extension. You introduce these experiments as "native-like", but I think it is important to preface this work with some discussion of how these simplified experiments are intended to test the distance requirement underlying the catalytic arrangement, and not the biological relevance of ubiquitin chain extension in trans (which appears to require viral scaffolding).

b. If the initial rounds of ubiquitin chain extension occur in trans as a safety against TRIM21 signaling activation in the absence of a scaffolding viral particle, then I wonder if you could test this model by bypassing this step. Would a linear fusion of ubiquitin molecules to the TRIM21 N-terminus approximate a K63-linked chain sufficiently well to induce chain extension in cis and signal activation in the absence of virus? If so, then your chain length requirement for the changeover to extension in cis could also be tested by fusing two, three, or four ubiquitin molecules to TRIM21.

c. A final model figure of how TRIM21 would oligomerize and satisfy the catalytic arrangement on the surface of a viral particle would make for a nice addition. The structural model in Supplementary Fig. 11 is useful as well, but difficult to interpret without a cartoon model alongside.

Minor comments:

1) The Ub-RING nomenclature are defined, but later a simpler Ub-R nomenclature is used that should either be defined or changed for consistency.

2) The statement in the introduction about E2s encoding linkage type and E3s selecting for substrate is oversimplified and should be rephrased to account for examples of linkage-specific E3s as well as substrate-specific E2s.

3) Units used for protein concentrations throughout the Methods section should be double-checked. For example, the pKa experiments list the concentration of ubiquitin as 250 mM, and in several instances the E1 concentration is also listed as mM.

4) I could not find a reference to Supplementary Figs. 9a and b in the text.

5) The concluding sentence of the Discussion section is a bit of a conceptual leap without additional logic to support it.

6) The figure legends describe stereo images on several occasions, but no stereo images are provided.

Reviewer #2 (Remarks to the Author):

The manuscript by Kiss et al (Leo James and coworkers) reports an important discovery regarding the assembly of ubiquitin chains and suggests a mechanism by which substrate binding regulates the E3 ligase activity of TRIM21.

TRIM21 is a well studied member of the TRIM family of ubiquitin E3 ligases that has an important role in immune responses because it recognises antibody-coated viruses and brings about their neutralisation. Prior studies have revealed that TRIM21 works in partnership with two E2s, Ube2W and Ubc13, to promote the addition of lysine-63 linked chains on its own N-terminus. Unusually, these K63-linked chains promote the destruction of antibody bound pathogens.

The James group have contributed significantly to our understanding of TRIM21 function and have previously reported the details of N-terminal ubiquitylation and the structure of a TRIM21 RING dimer bound to an E2~Ub conjugate. In the present study they bring this prior work together and report the structure of the RING domain of TRIM21 that is covalently linked to ubiquitin (mimicking the product of the Ube2W reaction) bound to a Ubc13~Ub conjugate. This structure, together with mutagenesis studies, provides insight into the assembly of K63 linked chains. The authors then go on to use a range of different TRIM21 constructs to suggest how substrate binding enhances activity and propose a model whereby addition of ubiquitin occurs 'in trans' and then 'in cis'.

Overall the work provides important insights into our understanding of ubiquitin chain assembly and the function of TRIM21.

Main points:

While the work is technically sound, the manuscript is written for a specialist audience and assumes considerable prior knowledge. The legends for many figures also include insufficient information that means interpretation is difficult.

The model for 'in trans' followed by 'in cis' activity is attractive, and the data appears consistent with this. However, this model is not directly tested. Can this be directly tested by using combinations of proteins with different mutations (i.e. E2-binding mutants and K0-ubiquitin)? Additional evidence to support this proposition would be valuable.

Minor points:

- i) Page 5/6 discussion about B factors of the b1-b2 loop – this could be extended as it is unclear exactly what the authors are suggesting.
- ii) Page 6, para starting line 130. Discussion of the Ubc13~Ub/Uev2 complex is included. It is unclear how this extends the prior work of Eddins et al.?
- iii) Page 6 discussion of the differences between the structure of Ubc13/Uev in the presence and absence of TRIM21 would be better illustrated by a close-up side-by-side comparison.
- iv) Page 7 a model is proposed whereby initial ubiquitin molecules are added in trans, followed by addition in cis. While subsequent experiments build on this model it is not directly proven.
- v) It may be helpful to move Figure 5a, or a version of it earlier and include the names of the constructs alongside the domains.
- vi) Page 8, line 190 and 199 – the authors use the term 'catalytic RING topology' here and in other places. The authors might wish to review this term. While the 'catalytic RING topology' is defined as including a RING dimer, there is some mismatch with the next figure/data. Also 'chain building complex' is referred to earlier and the distinction is easy to overlook.
- v) Fig 4 – it would be interesting to know if fusion of tetra ubiquitin to the N-terminus of the RBCCPS or RRCCPS constructs promotes ubiquitylation?
- vi) It is unclear how addition of antibodies stimulates TRIM activity in vitro.
- vii) Page 9, line 219 signals that free ubiquitin chains could be observed in Figure 4c, however no anti-ubiquitin blots are included in this figure.
- viii) Page 10, line 232, the 'Trim away' experiment will not be obvious to many, this should be briefly described and the use of 'Trim-Away' as a noun should be minimised.
- ix) It might be helpful if the authors encapsulated their model in a schematic.

Figure 1: Panel a is small and the inclusion of surface and ribbon for one molecule does not appear to enhance interpretation. The colours of the labels need to be revised as the pale colours are not visible when printed. Panel b is labelled as the canonical model but this is not very informative.

Likewise, panel c takes some deciphering and may be more usefully referred to as the Ubc13 catalytic complex or something similar.

In the end panels b&c are generated by combining elements from three different symmetry mates. It may be better to remove the extraneous components from panel c and just show the elements that make up the catalytic complex that becomes the focus of Figure 2.

Figure 2: Ub-R is not defined. The two blots included appear to have different amounts of ubiquitin/or have been exposed for different lengths. The legend or figure should also indicate that this is an anti-Ub western. Commassie stained gels should also be included (supplementary?) to demonstrate E2 loadings are comparable.

Figure 3: Panel a, the authors should consider whether all molecules included in the surface diagram are required. Panel b, it would be helpful to show the B factors plotted against residue number – as presented it is difficult to interpret. Panel c is difficult to see. Here the focus is on the modelled ubiquitin chains and it might be easier to see this if some of the other chains were shown as just a surface (or just a ribbon).

Supp Figure 3: It would be helpful if the molecules overlaid were described.

Supp Figure 4&5: Labels are hard to see and

Reviewer #3 (Remarks to the Author):

This is an exceptionally well-written paper, describing a careful and thorough and scientific investigation into the mechanism of RING-ubiquitination by RINGs using TRIM 21 as the model. The authors comprehensively demonstrate the role of D119 in catalysis, and perform elegant experiments to test the hypothesis that the topology they observe is the one required for substrate-anchored K63-ubiquitin chain elongation.

In the opinion of this reviewer, the identification of the catalytic base (D119) is a really clear result. Nothing in the manuscript is overinterpreted or stretched. I am fully supportive of publication, there is no need for any further experimentation to support the claims.

I have a couple of minor points the authors could consider to help the general audience - in the introduction they assert that the TRIM family contains the greatest variety of RINGs, but I am not sure what is meant by that. Are there different types of RINGs themselves? Are they referring to biochemical type, or to biological pathway?

I would also consider citing the study from Maria Sunnerhagen's group last year that found the equivalent aspartate in Ube2E1 (fig 6, Anandapadamanaban et al., JBC, 2019) to be important for deprotonating the incoming lysine.

On page 4, they refer to a 2.2Å structure as high-resolution - I think this is high-res in terms of macromolecules but possibly not in terms of chemical mechanism. I recommend just stating the resolution and let the reader put the judgement on it.

REVIEWER COMMENTS

We thank the reviewers for their thorough analysis of our work.

Reviewer #1 (Remarks to the Author):

In their latest work, Kiss et al. report a crystal structure that captures TRIM21 in the act of building a polyubiquitin chain onto its N-terminus and define this as the catalytic arrangement that is required for signaling in response to viral detection. From a structural standpoint, this work builds upon previous work that has described individual elements such as E2 activation and polyubiquitin chain formation, but is the first to visualize the act of ubiquitin chain extension onto a “primed” monoubiquitinated substrate. From their structure, the authors observe interactions in the E2 catalytic center that activate the incoming substrate lysine. This interaction has previously been tested in other systems of ubiquitin conjugation, but the authors do a nice job of demonstrating its importance for chain assembly in their system. The bulk of the remaining work focuses on formation of the so-called “catalytic arrangement”, which sets strict distance parameters that are required for extension of the first few ubiquitin molecules in *trans*, after which the chain can be rapidly extended through additions in *cis*.

The work is of high quality and is described very clearly. As the authors point out, the experiments addressing activation of the incoming substrate lysine confirm a body of previous work that propose this mechanism. Perhaps the most striking result is the mechanism underlying formation of the catalytic arrangement. This model could be explained more clearly with additional discussion in the text and perhaps a cartoons schematic for a final figure, but the underlying principle is that the initial extension of self-anchored ubiquitin chains can only occur in *trans* if TRIM21 molecules are assembled onto a viral particle. Once the catalytic arrangement can be formed through this virus-induced oligomerization, ubiquitin chain extension can begin in *trans* and, after a certain threshold, can rapidly continue in *cis* to produce a K63-linked ubiquitin signal. This represents a new and interesting mechanism of ubiquitin signaling regulation. With some revisions to strengthen the communication of this model, I recommend publication.

We thank this reviewer for the thorough analysis of our work and for the strongly positive comments.

Major comments:

1) Though the kinetic experiments presented in Figure 2 and Supplementary Figure 2 are well thought-out and internally consistent, there are several points that should be addressed to justify certain aspects of the methodology:

a. Is there any reason to believe that the auto-ubiquitinated form of Ube2N would not display altered rates of ubiquitin chain assembly? Is this accounted for in some way?

The additional Ube2N~Ub₂ band was a result we did not fully expect when establishing these kinetic experiments. It most likely arises from a side reaction with one of the lysine residues close to the active site. One way to circumvent this would have been to mutate such residues (e.g. K92) as we have done for the isopeptide charging. However, we decided against this as our aim was to study the native active site. This band comprises 5-10 % of the Ube2N~Ub signal. In addition, Ube2N~Ub₂ appears to discharge ubiquitin at rates similar to Ube2N~Ub (see below) in Michaelis Menten kinetics of Ube2N-WT. In contrast to data shown in Supplementary Fig. 2, here we show the molar amount of Ube2N~Ub and Ube2N~Ub₂ in nM against the concentration of UbΔGG.

We therefore concluded it was reasonable to assume that all Ube2N~Ub species probably behave similarly within the conditions of our assay. All kinetic parameters were derived from analysis of the amount of formed diUb alone. We have added a note in the Methods section, explaining that the Ube2N~Ub₂ species is a side product of the charging and appears to behave similarly to Ube2N~Ub.

b. The His-Ub is presented as a dedicated donor ubiquitin and the UbΔGG as a dedicated acceptor, but without incorporating a K63R mutation into the His-Ub then I don't see how it can truly only act as a donor. I recognize that the concentration of His-Ub is much lower than Ub-ΔGG in the final chain assembly reaction, but formally it should still be possible for His-Ub to act as an acceptor. In fact, for some of the slower reactions (e.g. Supplementary Figure 2C, D119A), a higher molecular

weight di-ubiquitin band is visible that could reflect assembly of two His-Ub molecules. I don't think this could be affecting the results but it should be discussed.

The reviewer is completely right. Theoretically, His-Ub can act as donor and acceptor ubiquitin, but as pointed out by the reviewer, the much higher concentration of Ub Δ GG substantially disfavours the event of His-Ub acting as acceptor (e.g. in pK_s measurements the reaction contains 3 μ M His-Ub and 250 μ M Ub Δ GG). However, for the slowest mutant D119A such a band can indeed be identified in some of the slower conditions. Since this reflects a rather rare event at low levels, and in agreement with this reviewer, we conclude that this effect does not affect our results. Nonetheless, we extended our introduction into the assay in the Methods section, to clarify the exact design of these experiments.

c. For each panel in Supplementary Figure 2, the reaction timepoint selected for analysis should be listed.

This information has been added to the Methods section.

2) As the more innovative product of this work, the mechanism of viral-scaffolded oligomerization that leads to the catalytic arrangement for chain extension deserves more focus.

a. The data presented in Figures 4 and 5 are compelling and nicely speak to your model for a catalytic arrangement requirement. Until I read about TRIM21 oligomerization on a viral scaffold in the discussion, however, I was left wondering what the biological relevance of your structure was if Fc-bound full-length TRIM21 could not activate chain extension. You introduce these experiments as "native-like", but I think it is important to preface this work with some discussion of how these simplified experiments are intended to test the distance requirement underlying the catalytic arrangement, and not the biological relevance of ubiquitin chain extension *in trans* (which appears to require viral scaffolding).

We thank the referee for this helpful suggestion. We have changed the results section, no longer introducing the Fc-induced ubiquitination assays as native-like, but rather as a tool to test our hypothesis. In addition, we have extended our discussion about how the catalytic RING topology is established on the virus, including a new Figure 6, presenting a model of the mechanism.

b. If the initial rounds of ubiquitin chain extension occur in *trans* as a safety against TRIM21 signaling

activation in the absence of a scaffolding viral particle, then I wonder if you could test this model by bypassing this step. Would a linear fusion of ubiquitin molecules to the TRIM21 N-terminus approximate a K63-linked chain sufficiently well to induce chain extension in *cis* and signal activation in the absence of virus? If so, then your chain length requirement for the changeover to extension in *cis* could also be tested by fusing two, three, or four ubiquitin molecules to TRIM21.

We thank the reviewer warmly for suggesting this very highly relevant and, as it turns out, successful experiment. We have generated Ub_n-R-R-PS constructs containing one to four linearly fused ubiquitin molecules (see new Fig. 4). Linear and K63-linked chains are structurally sufficiently similar¹ to be used for this purpose. We tested the following constructs in our Fc-induced ubiquitination experiment: Ub₁-R-R-PS, Ub₂-R-R-PS, Ub₃-R-R-PS, Ub₄-R-R-PS. TRIM21 constructs carrying one to three depend on addition of Fc for self-ubiquitination. However, the construct carrying four ubiquitin molecules, is highly active in absence and presence of Fc. This proves that with a sufficiently long ubiquitin chain the ubiquitination reaction switches to a highly efficient *cis* mechanism. The less *trans* ubiquitination is required, the more efficient the reactions become, as can be seen by the band of the Ub_n-R-R-PS constructs in Fig. 4b. Of note, in the case of Ub₄-R-R-PS, nearly all of the protein gets converted into heavily ubiquitin-conjugated species.

c. A final model figure of how TRIM21 would oligomerize and satisfy the catalytic arrangement on the surface of a viral particle would make for a nice addition. The structural model in Supplementary Fig. 11 is useful as well, but difficult to interpret without a cartoon model alongside.

We have designed a new model now shown in Fig. 6, explaining how TRIM21 and TRIM5 oligomerization enables the catalytic RING topology. In addition, we have extended the discussion section to explain this in further detail.

Minor comments:

1) The Ub-RING nomenclature are defined, but later a simpler Ub-R nomenclature is used that should either be defined or changed for consistency.

We have changed the paper to make consistent use of the Ub-R nomenclature.

2) The statement in the introduction about E2s encoding linkage type and E3s selecting for substrate

is oversimplified and should be rephrased to account for examples of linkage-specific E3s as well as substrate-specific E2s.

We agree with the reviewer; we felt the best solution was to remove this sentence.

3) Units used for protein concentrations throughout the Methods section should be double-checked. For example, the pK_a experiments list the concentration of ubiquitin as 250 mM, and in several instances the E1 concentration is also listed as mM.

We thank the reviewer for pointing out these oversights. We have carefully checked the whole manuscript and have corrected such issues.

4) I could not find a reference to Supplementary Figs. 9a and b in the text.

We have corrected the manuscript such that all Figures and Supplementary Figures are now referenced within the text.

5) The concluding sentence of the Discussion section is a bit of a conceptual leap without additional logic to support it.

We have extended our concluding sentence with additional information. In the past, many ligases have been shown to assemble into larger complexes. Among them are many that use Ube2N for the formation of K63-linked ubiquitin chains, such as TRAF6² or RIPLET³. TRAF6 is particularly interesting since its RINGs dimerize but its coiled-coil is a trimer, so arrangements conceptually similar to TRIM21 or TRIM5 could therefore be formed². We thus suggest that the mechanism shown here might be used by other such ligases as well.

6) The figure legends describe stereo images on several occasions, but no stereo images are provided.

We apologise for this error on our part. These figure legends have been updated accordingly and a stereo image has been added as Supplementary Fig. 1b.

Reviewer #2 (Remarks to the Author):

The manuscript by Kiss et al (Leo James and co-workers) reports an important discovery regarding the assembly of ubiquitin chains and suggests a mechanism by which substrate binding regulates the E3 ligase activity of TRIM21.

TRIM21 is a well studied member of the TRIM family of ubiquitin E3 ligases that has an important role in immune responses because it recognises antibody-coated viruses and brings about their neutralisation. Prior studies have revealed that TRIM21 works in partnership with two E2s, Ube2W and Ubc13, to promote the addition of lysine-63 linked chains on its own N-terminus. Unusually, these K63-linked chains promote the destruction of antibody bound pathogens.

The James group have contributed significantly to our understanding of TRIM21 function and have previously reported the details of N-terminal ubiquitylation and the structure of a TRIM21 RING dimer bound to an E2~Ub conjugate. In the present study they bring this prior work together and report the structure of the RING domain of TRIM21 that is covalently linked to ubiquitin (mimicking the product of the Ube2W reaction) bound to a Ubc13~Ub conjugate. This structure, together with mutagenesis studies, provides insight into the assembly of K63 linked chains. The authors then go on to use a range of different TRIM21 constructs to suggest how substrate binding enhances activity and propose a model whereby addition of ubiquitin occurs 'in *trans*' and then 'in *cis*'.

Overall the work provides important insights into our understanding of ubiquitin chain assembly and the function of TRIM21.

We thank this reviewer for the thorough analysis of our work and for the strongly positive comments.

Main points:

While the work is technically sound, the manuscript is written for a specialist audience and assumes considerable prior knowledge. The legends for many figures also include insufficient information that means interpretation is difficult.

We have edited the manuscript and Figure legends to make them more accessible to readers.

The model for '*in trans*' followed by '*in cis*' activity is attractive, and the data appears consistent with this. However, this model is not directly tested. Can this be directly tested by using combinations of proteins with different mutations (i.e. E2-binding mutants and KO-ubiquitin)? Additional evidence to support this proposition would be valuable.

We thank the reviewer for raising this point. We have performed additional experiments to prove our model where TRIM21 self-ubiquitination is initiated in *trans*, and then switches into a more efficient *cis* mechanism. This extremely helpful experiment was suggested both by this reviewer (Minor points, v) and reviewer 1 (Main points, 2, b), and we thank both reviewers warmly for the suggestion. We have generated Ub_n-R-R-PS constructs containing one to four linearly fused ubiquitin molecules (see new Fig. 4). Linear and K63-linked chains are sufficiently similar structurally¹ to allow the linear chains be used for this purpose. We tested the following constructs in our Fc-induced ubiquitination experiment: Ub₁-R-R-PS, Ub₂-R-R-PS, Ub₃-R-R-PS, Ub₄-R-R-PS. TRIM21 constructs carrying one to three ubiquitins depend on addition of Fc for self-ubiquitination. However, the construct carrying four ubiquitin molecules is highly active in either the absence or the presence of Fc. This proves that with a sufficiently long ubiquitin chain the ubiquitination reaction switches to a highly efficient *cis* mechanism. The less *trans* ubiquitination is required, the more efficient the reactions become, as can be seen by comparing the bands of the different Ub_n-R-R-PS constructs in Fig. 4b. Of note, in the case of Ub₄-R-R-PS, nearly all of the protein gets converted into heavily ubiquitin-conjugated species.

Minor points:

i) Page 5/6 discussion about B factors of the b₁-b₂ loop – this could be extended as it is unclear exactly what the authors are suggesting.

We understand and appreciate the reviewer's point. However, we were intrigued to see the differences in the B factors between donor and acceptor ubiquitin, and although we cannot as yet provide an interpretation we wanted to share this finding, as we believe the information might be useful to other ubiquitin aficionados.

ii) Page 6, para starting line 130. Discussion of the Ubc13~Ub/Uev2 complex is included. It is unclear how this extends the prior work of Eddins et al.?

We kept the discussion of our Ube2N~Ub:Ube2V2 complex relatively short since the overall conclusions from this structure are similar to those in the work on yeast enzymes by ⁴, as stated in the main text. However, we still included and introduced the structure as it is a complex of human proteins, which we felt makes it a more suitable comparator for our Ub-R:Ube2N~Ub:Ube2V2 structure than a complex of yeast proteins would have been.

iii) Page 6 discussion of the differences between the structure of Ubc13/UeV in the presence and absence of TRIM21 would be better illustrated by a close-up side-by-side comparison.

We have added such a Figure as Supplementary Fig. 5c.

iv) Page 7 a model is proposed whereby initial ubiquitin molecules are added in *trans*, followed by addition in *cis*. While subsequent experiments build on this model it is not directly proven.

As described above (this reviewer (main points, 2) and reviewer 1 (main points, 2, b), we now provide new evidence strongly supporting this model.

v) It may be helpful to move Figure 5a, or a version of it earlier and include the names of the constructs alongside the domains.

We agree with the reviewer and have changed Figures 4 and 5 accordingly.

vi) Page 8, line 190 and 199 – the authors use the term ‘catalytic RING topology’ here and in other places. The authors might wish to review this term. While the ‘catalytic RING topology’ is defined as including a RING dimer, there is some mismatch with the next figure/data. Also ‘chain building complex’ is referred to earlier and the distinction is easy to overlook.

We introduce the term catalytic RING topology the following way in the results section on p7: “We refer to this arrangement as the ‘catalytic RING topology’, where a RING dimer acts as enzyme reacting with at least one further RING as substrate for ubiquitination.” In the Figure, the ‘substrate’ we show is a Ub-R, and we also show the next Ub-R with which it forms a dimer. We decided to do this, as we wanted the crystal structure Figures to be as consistent as possible for readers who are

not specialists in structural biology. In addition, we explain this concept further in the discussion, where we also have a new model Figure (Fig. 6).

We have removed the term 'chain building complex' for clarity.

v) Fig 4 – it would be interesting to know if fusion of tetra ubiquitin to the N-terminus of the RBCCPS or RRBCCPS constructs promotes ubiquitylation?

We thank the reviewer for this suggestion. As explained above (this reviewer, main points, 2) and reviewer 1 (main points, 2, b) we have now carried out these experiments. However, we did not do this with R-R-B-CC-PS, but rather R-R-PS. Unfortunately, expressing, purifying and working with full-length TRIM21 protein is extremely complicated as this protein behaves very poorly. Producing enough Ub-R-(R-)B-CC-PS was already challenging for the assays presented in Figure 3d and Supplementary Fig. 9. Nonetheless, we believe the constructs that we used do answer the reviewer's question fully, as fusing a tetra-ubiquitin to TRIM21 R-R-PS makes it constitutively active (see Fig. 4b).

vi) It is unclear how addition of antibodies stimulates TRIM activity *in vitro*.

Binding of antibody (or Fc) does not strictly activate full length TRIM21 *in vitro*, as can be seen in Figure 3. Nonetheless, we designed these experiments in the way that we did because in the constructs lacking B-box and coiled-coil we can use the addition of Fc to bring RING domains into proximity, enabling formation of the catalytic RING topology. We tried to convey this in our cartoon sketches in Figure 3 and show additional structural models in Supplementary Fig. 8.

When restricting incoming antibody-coated viruses (or other agents such as proteins in Trim-Away), the catalytic RING topology is induced by higher order clustering of multiple TRIM21:antibody complexes on the target. To explain this more clearly, we have expanded our discussion and provide a new model Figure (Fig. 6), which shows this process in detail.

vii) Page 9, line 219 signals that free ubiquitin chains could be observed in Figure 4c, however no anti-ubiquitin blots are included in this figure.

We apologise for the confusion. The blot showing free ubiquitin is shown in Supplementary Fig. 9c. While the upper signals most likely represent ubiquitin that is conjugated to TRIM21, signals below

the Ub-TRIM21 bands can only originate from free ubiquitin chains. Here, we do not observe free ubiquitin chains longer than di-ubiquitin (and potentially very faint bands representing tri-ubiquitin). We have updated the text accordingly.

viii) Page 10, line 232, the 'Trim away' experiment will not be obvious to many, this should be briefly described and the use of 'Trim-Away' as a noun should be minimised.

We thank the reviewer for pointing this out. We have changed our wording to make clear that a Trim-Away experiment is a targeted protein degradation experiment and also we have tried to avoid using it as a noun.

ix) It might be helpful if the authors encapsulated their model in a schematic.

We fully agree with the reviewer and have provided a new Figure with a model (Fig. 6).

Figure 1: Panel a is small and the inclusion of surface and ribbon for one molecule does not appear to enhance interpretation. The colours of the labels need to be revised as the pale colours are not visible when printed. Panel b is labelled as the canonical model but this is not very informative. Likewise, panel c takes some deciphering and may be more usefully referred to as the Ubc13 catalytic complex or something similar.

In the end panels b&c are generated by combining elements from three different symmetry mates. It may be better to remove the extraneous components from panel c and just show the elements that make up the catalytic complex that becomes the focus of Figure 2.

We increased the size of the Figure and changed the colours of the labels to make them clearer when printed. The reason for including both surface and ribbon is to explain how we generated our structure. As the reviewer also pointed out, the 'canonical model' contains data from three different symmetry mates, making its origin rather complicated to conceptualise. We hope the modified version of this figure will convey our message more clearly.

Figure 2: Ub-R is not defined. The two blots included appear to have different amounts of ubiquitin/or have been exposed for different lengths. The legend or figure should also indicate that this is an anti-Ub western. Commassie stained gels should also be included (supplementary?) to demonstrate E2 loadings are comparable.

We agree with the reviewer. We have therefore provided another assay in Supplementary Fig. 3a, where every blot contains a Ube2N-WT control. As the concentration of E2 enzyme in the assay is rather low, it cannot be seen on a Coomassie gel. Instead, we have run the stock solutions (10 μ M), that were used for the assay, in an LDS-PAGE and provide this gel as Supplementary Fig. 3b.

Figure 3: Panel a, the authors should consider whether all molecules included in the surface diagram are required.

We agree with the reviewer that some readers might be confused that the 'substrate' RING here is a dimer as well. Since this is a crystal structure, all RINGS are identical and were generated via symmetry operations. We decided to be consistent with the way we present the structure to make it easier for the non-structural biology expert readers to understand our structure when downloading it from the PDB.

Panel b, it would be helpful to show the B factors plotted against residue number – as presented it is difficult to interpret.

While revising our manuscript, we have changed Figures 3 and 4 significantly. The original content of panel b is not part of the Figure anymore. However, the B-factors are plotted against residue number in Supplementary Fig. 1c.

Panel c is difficult to see. Here the focus is on the modelled ubiquitin chains and it might be easier to see this if some of the other chains were shown as just a surface (or just a ribbon).

We thank the reviewer for this suggestion. What was Fig. 3c is now Fig. 4a and we now show Ube2N~Ub:Ube2V2 as surface while showing Ub₄-R and Ub-R as cartoon, to make the Figure easier to interpret.

Supp Figure 3: It would be helpful if the molecules overlaid were described.

We have added further descriptions to the Figure legend.

Supp Figure 4&5: Labels are hard to see and

We have changed the labels to make them easier to see.

Reviewer #3 (Remarks to the Author):

This is an exceptionally well-written paper, describing a careful and thorough and scientific investigation into the mechanism of RING-ubiquitination by RINGs using TRIM21 as the model. The authors comprehensively demonstrate the role of D119 in catalysis, and perform elegant experiments to test the hypothesis that the topology they observe is the one required for substrate-anchored K63-ubiquitin chain elongation.

In the opinion of this reviewer, the identification of the catalytic base (D119) is a really clear result. Nothing in the manuscript is overinterpreted or stretched. I am fully supportive of publication, there is no need for any further experimentation to support the claims.

We thank the reviewer for the strongly positive comments.

I have a couple of minor points the authors could consider to help the general audience - in the introduction they assert that the TRIM family contains the greatest variety of RINGs, but I am not sure what is meant by that. Are there different types of RINGs themselves? Are they referring to biochemical type, or to biological pathway?

We thank the reviewer for pointing out the imprecision of this sentence. What we intended to say was that no other protein family contains as many RINGs as do TRIM proteins. While considering all the different complexes formed by cullin-RING ligases, they still only contain 2 different RING domains (Rbx1 and 2). On the other hand, the TRIM ligases comprise ~100 different proteins⁵, each carrying a different RING domain. We have edited the text to make this point more clearly.

I would also consider citing the study from Maria Sunnerhagen's group last year that found the equivalent aspartate in Ube2E1 (fig 6, Anandapadamanaban et al., JBC, 2019) to be important for deprotonating the incoming lysine.

We carefully read this nice work from Maria Sunnerhagen's group⁶. While the equivalent aspartate (and another one nearby) were studied in detail and shown to be important for ubiquitination, we

could not find any mention of deprotonation in this work. However, others suggested this role already for Ube2D1⁷ and for the SUMO E2 Ube2I the equivalent residue has been shown to be important for deprotonation⁸. These works are referenced in our article.

On page 4, they refer to a 2.2Å structure as high-resolution - I think this is high-res in terms of macromolecules but possibly not in terms of chemical mechanism. I recommend just stating the resolution and let the reader put the judgement on it.

We fully agree with the reviewer and have changed our manuscript accordingly.

- 1 Komander, D. *et al.* Molecular discrimination of structurally equivalent Lys 63-linked and linear polyubiquitin chains. *EMBO Rep* **10**, 466-473, doi:10.1038/embor.2009.55 (2009).
- 2 Napetschnig, J. & Wu, H. Molecular basis of NF-kappaB signaling. *Annu Rev Biophys* **42**, 443-468, doi:10.1146/annurev-biophys-083012-130338 (2013).
- 3 Cadena, C. *et al.* Ubiquitin-Dependent and -Independent Roles of E3 Ligase RIPLET in Innate Immunity. *Cell* **177**, 1187-1200 e1116, doi:10.1016/j.cell.2019.03.017 (2019).
- 4 Eddins, M. J., Carlile, C. M., Gomez, K. M., Pickart, C. M. & Wolberger, C. Mms2-Ubc13 covalently bound to ubiquitin reveals the structural basis of linkage-specific polyubiquitin chain formation. *Nat Struct Mol Biol* **13**, 915-920, doi:10.1038/nsmb1148 (2006).
- 5 Marin, I. Origin and diversification of TRIM ubiquitin ligases. *PLoS One* **7**, e50030, doi:10.1371/journal.pone.0050030 (2012).
- 6 Anandapadamanaban, M. *et al.* E3 ubiquitin-protein ligase TRIM21-mediated lysine capture by UBE2E1 reveals substrate-targeting mode of a ubiquitin-conjugating E2. *J Biol Chem* **294**, 11404-11419, doi:10.1074/jbc.RA119.008485 (2019).
- 7 Plechanovova, A., Jaffray, E. G., Tatham, M. H., Naismith, J. H. & Hay, R. T. Structure of a RING E3 ligase and ubiquitin-loaded E2 primed for catalysis. *Nature* **489**, 115-120, doi:10.1038/nature11376 (2012).
- 8 Yunus, A. A. & Lima, C. D. Lysine activation and functional analysis of E2-mediated conjugation in the SUMO pathway. *Nat Struct Mol Biol* **13**, 491-499, doi:10.1038/nsmb1104 (2006).

REVIEWERS' COMMENTS

Reviewer #1 (Remarks to the Author):

The authors have nicely addressed all of my comments. Both the presentation and validation of their mechanism for ubiquitin chain extension are greatly improved. I look forward to seeing this work in publication.

Sincerely,

Jonathan Pruneda

Reviewer #2 (Remarks to the Author):

The authors have done a good job of addressing my concerns, as well as those of the other referees. In my view the manuscript is considerably improved in that it is more accessible to a wider audience and the additional data included in the new Figure 4b is an excellent addition that considerably strengthens the trans/cis proposal.

The model shown in Figure 6 is a good addition but I wonder if the authors can highlight the location of the RING domains in panels a&c? Perhaps even with a circle that leads to the expanded view below. Also in the expanded view in panels b&d it may be possible to improve the clarity of what is shown - I suspect that the current version will leave most readers somewhat confused!

With that small caveat, I am happy for the manuscript to be published.